# The ubiquitination landscape of the influenza A virus polymerase

Franziska Günl [1,2], Tim Krischuns [1,15], Julian A. Schreiber [3,4], Lea Henschel[1], Marius Wahrenburg[1,5], Hannes C. A. Drexler [6], Sebastian A. Leidel[7,8], Vlad Cojocaru [9,10,11], Guiscard Seebohm [4], Alexander Mellmann [2,12], Martin Schwemmle [13,14], Stephan Ludwig [1,2] & Linda Brunotte [1,2] ✉

During influenza A virus (IAV) infections, viral proteins are targeted by cellular E3 ligases for modification with ubiquitin. Here, we decipher and functionally explore the ubiquitination landscape of the IAV polymerase proteins during infection of human alveolar epithelial cells by applying mass spectrometry analysis of immuno-purified K-ε-GG (di-glycyl)-remnant-bearing peptides. We have identified 59 modified lysines across the three subunits, PB2, PB1 and PA of the viral polymerase of which 17 distinctively affect mRNA transcription, vRNA replication and the generation of recombinant viruses via non-proteolytic mechanisms. Moreover, further functional and in silico analysis indicate that ubiquitination at K578 in the PB1 thumb domain is mechanistically linked to dynamic structural transitions of the viral polymerase that are required for vRNA replication. Mutations K578A and K578R differentially affect the generation of recombinant viruses by impeding cRNA and vRNA synthesis, NP binding as well as polymerase dimerization. Collectively, our results demonstrate that the ubiquitin-mediated charge neutralization at PB1-K578 disrupts the interaction to an unstructured loop in the PB2 N-terminus that is required to coordinate polymerase dimerization and facilitate vRNA replication. This provides evidence that IAV exploits the cellular ubiquitin system to modulate the activity of the viral polymerase for viral replication.

Influenza A viruses (IAV) are respiratory pathogens of the *Orthomyxoviridae* family that pose a substantial threat to global health through seasonal epidemics and recurring pandemics. The IAV genome consists of eight segments of negative-sense single-strand RNA (vRNA).

Each vRNA is encapsidated by multiple copies of the nucleoprotein (NP) and one copy of the trimeric viral RNA-dependent RNA polymerase (RdRP), composed of the subunits PB2, PB1, and PA[1,2] to form viral ribonucleoprotein (vRNPs) complexes[3]. Transcription of viral

[1]Institute of Virology Muenster, University of Muenster, Muenster, Germany. [2]Interdisciplinary Center for Clinical Research (IZKF), University of Muenster, Muenster, Germany. [3]Institute for Pharmaceutical and Medicinal Chemistry, University of Muenster, Muenster, Germany. [4]Cellular Electrophysiology and Molecular Biology, Institute for Genetics of Heart Diseases (IfGH), Department of Cardiovascular Medicine, University Hospital Muenster, Muenster, Germany. [5]Department of Orthopedic Oncology, University Hospital Essen, Essen, Germany. [6]Bioanalytical Mass Spectrometry, Max Planck Institute for Molecular Biomedicine, Muenster, Germany. [7]Department of Chemistry, Biochemistry and Pharmaceutical Sciences, University of Bern, Bern, Switzerland. [8]Multidisciplinary Center for Infectious Diseases, University of Bern, Bern, Switzerland. [9]Max Planck Institute for Molecular Biomedicine, Muenster, Germany. [10]Computational Structural Biology Group, Utrecht University, Utrecht, The Netherlands. [11]STAR-UBB Institute, Babeş-Bolyai University, Cluj-Napoca, Romania. [12]Institute of Hygiene, University Hospital Muenster, University of Muenster, Muenster, Germany. [13]Institute of Virology, University Medical Center Freiburg, Freiburg, Germany. [14]Faculty of Medicine, University of Freiburg, Freiburg, Germany. [15]Present address: Institut Pasteur, Université Paris Cité, CNRS UMR3569, Unité Biologie des ARN et Virus Influenza, Paris, France. ✉e-mail: brunotte@uni-muenster.de

mRNAs and replication of the viral genome is mediated by RdRPs and occurs in the cell nucleus. While incoming vRNPs are capable to directly perform mRNA synthesis following cap-snatching from host-derived pre-mRNAs[4–6], vRNA replication depends on newly produced viral proteins and polymerase complexes. It proceeds in a primer-independent manner that involves the generation of full-length positive-sense complementary RNA (cRNA) intermediates, which serve as templates for vRNA synthesis[7].

Structural investigations revealed that the IAV RdRP is a flexible and highly dynamic protein complex that adopts different conformations during mRNA transcription and vRNA replication[5,8]. The current model for vRNA replication suggests that the transition from the transcriptase to the replicase is promoted by the interaction with a second viral polymerase forming an asymmetric dimer[9,10]. Moreover, the formation of the asymmetric dimer facilitates encapsidation of the nascent cRNA strand. In the second step of genome replication, initiation of vRNA synthesis from cRNA templates requires formation of a symmetric polymerase dimer through interaction with a trans-activating polymerase[11,12]. To maintain the optimal balance between mRNA transcription and vRNA replication during the viral life cycle chronologic assembly of the two dimers is critical and is suggested to be influenced by viral as well as cellular factors. In particular, host-mediated post-translational modifications (PTM) of viral proteins were proposed as dynamic molecular switches to fine-tune the actions of the RdRP[13,14]. Indeed, phosphorylation[15–19], ADP-ribosylation or acetylation[19–21] as well as linkage of ubiquitin (UB)[22–31] and ubiquitin-like modifiers (UBL), such as SUMO 1–3[32,33], NEDD8[34], or ISG15[35] to the polymerase subunits and NP have been reported. Among these modifiers, ubiquitination is the most abundant in human cells and regulates the functionality and longevity of a wide range of proteins and has been described for all subunits of the IAV RdRP with both pro- and antiviral outcomes. Conjugation of PTMs such as UB confers several effects. It introduces a binding surface for other UBL-binding proteins, but can also shield protein binding sites on the modified protein. In addition, UB transfers biological signals, e.g., the degradation by the cellular proteasome or translocation to a different cellular compartment. A less reported consequence of UB-linkage is the neutralization of the positive charge in the side chain of the acceptor lysine by formation of the dipeptide bond, which can result in structural alterations and affect interactions with other proteins[36]. Until today, the biological impact of site-specific modifications in the IAV polymerase proteins for the processes of viral mRNA transcription and genome replication has remained largely unknown.

In this study, we have performed mass spectrometry (MS) analyses of immuno-selected, K-ε-GG-(di-glycyl)-enriched viral proteins purified from infected human lung epithelial cells to unravel the ubiquitination landscape of the IAV polymerase during infection. We have identified 59 modified lysines across the subunits of the IAV RdRP, with UB being the most abundant UBL. Mutational analysis and generation of recombinant viruses by reverse genetics reveal that ubiquitination at position PB1-K578 in the PB1-thumb domain regulates the spatiotemporal positioning of an unstructured N-terminal loop in the PB2 subunit and thereby participates in the coordination of polymerase dimerization, NP binding and vRNA replication.

## Results

### Unraveling the ubiquitination landscape of the IAV polymerase during infection

Western blot analysis of co-precipitated strep-tagged RdRP subunits, PB2, PB1, and PA, with co-expressed UBLs revealed abundant UB modifications on all three subunits (Fig. 1a and Supplementary Fig. 1), which is in line with previous reports[27]. In contrast, NEDD8 was only detected on PA, while ISG15 was absent for all three subunits (Fig. 1b, c).

To determine the specific location of the UB/UBL-modified lysines within the IAV polymerase during the early phase of the viral infection cycle in human alveolar epithelial cells, we infected A549 cells with the human IAV strain A/WSN/1933 (H1N1, WSN) at a multiplicity of infection (MOI) of 20 for 5 h, followed by immunoselection of trypsin digested peptides harboring di-glycyl remnants and MS analysis (Fig. 1d, e).

In total, we identified 59 lysines with di-glycyl remnants distributed across all three subunits of the polymerase, of which 22 were located in PA and PB2, each, and 15 in PB1 (Table 1). The majority of the modified lysines are highly conserved among IAV strains from birds, swine and humans (Fig. 1f–h). To assess surface accessibility and spatial distribution of the modified lysines within the described conformational states, we generated sequence-adapted three-dimensional structure models of the trimeric WSN RdRP bound to promoters of vRNA (Fig. 2a) or cRNA (Fig. 2b) by homology modeling[8,9,11,37]. The majority of the modified lysines were exposed on the surface of the polymerase trimer in at least one of the conformations, suggesting that modification by E3 ligases can occur in the context of the individual proteins as well as the active polymerase trimer (summarized in Table 1). In PB1 some lysines were located within the inaccessible hydrophobic core, suggesting that modification occurred prior to the assembly of the polymerase trimer. Importantly, many lysines were located in functionally described subdomains of the polymerase (Fig. 2c). In PB2, modified lysines resided in two main clusters: the N-terminus, which extends to the N2 domain, and the combined C-terminal 627-NLD domain. Both regions are crucial for interactions with NP, PA, and PB1[38–45]. Moreover, some modified lysines were located in regions implicated in polymerase dimerization[9,11]. Only three modified lysines were detected in the flexible and exposed mid, cap-binding (CapB), and linker domains. Several modified lysines were embedded in nuclear localization signals (NLS) or mediated interaction with the host nuclear import protein importin-α, suggesting a putative function for UB/UBL modifications in nuclear-cytoplasmic shuttling of PB2 proteins and the RdRP complex[46,47].

In PA, seven di-glycyl-carrying lysines resided in the N-terminal endonuclease domain (Fig. 2c, middle panel), also including the catalytic residue K134[48–50]. Interestingly, four lysines directly faced toward the PB2-CapB domain in the transcriptase conformation of the trimeric polymerase, including K102 and K104, which were reported to interact with the 5′-capped mRNA-primer during transcription[50], suggesting that the cap-snatching process is a putative target for regulation by UB/UBL modification. Three modified residues were located in the PA linker region. Within the PA C-terminus three modified lysines were found in the region involved in the formation of the symmetric dimer[9,11,12] and six within the interaction site to the cellular RNA polymerase II (pol-II)[42,51]. In both, PA and PB1, several modified lysines were located in close proximity to the binding pocket of the 5′ hook of the vRNA and cRNA templates[42,52–55].

The central polymerase subunit PB1 exhibited the lowest number of modified lysines in our analysis. Furthermore, only 7 out of 15 lysines were surface exposed, while the rest was located in the hydrophobic core or covered by the other subunits within the trimeric complex (Fig. 2a, b, lower panel). Residue K480 is located at the rim of the NTP tunnel, where it has been shown to participate in NTP binding during RNA synthesis[42,55]. K578 resides in the PB1 thumb domain, which is a highly flexible subdomain that is part of the vRNA template exit tunnel[5]. In addition, it directly points toward the PB2-N1-domain and is located in close proximity to the interface of the symmetric dimer as well as the secondary binding site of the cRNA and vRNA 3′ end[5,11,56]. In summary, our results provide the first comprehensive map of UB/UBL modifications that are acquired across all subunits of the IAV viral polymerase at 5 h p.i. of human lung epithelial cells.

### Substitution of modified lysines impacts polymerase functions

To assess the functional impact of the site-specific modifications identified in our analysis, we substituted each lysine with a non-charged alanine (A) or a positively charged arginine (R) and

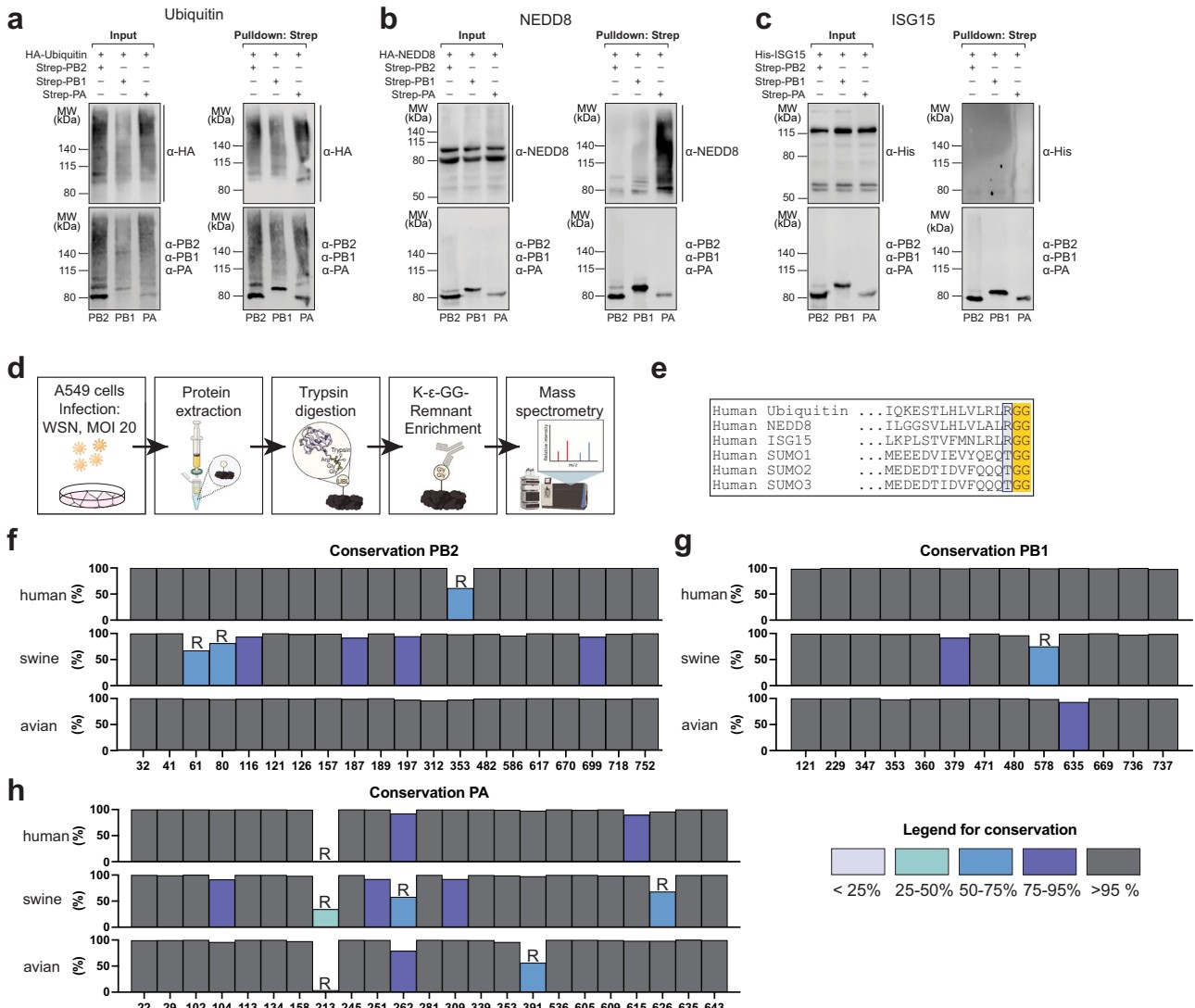

**Fig. 1 | Identification of site-specific UB modifications in the IAV polymerase.** **a–c** Western blot analysis of ubiquitin (UB), NEDD8 and ISG15 modification of the IAV polymerase subunits. A549 (UB) or HEK293-T (NEDD8, ISG15) were transfected with expression plasmids for strep-tagged polymerase subunits (PB2, PB1, or PA), respectively, together with plasmids for the expression of ubiquitin-like modifiers (UBLs) bearing an HA- or His-tag. UBL-modified polymerase subunits were precipitated under denaturing conditions using Strep-Tactin® bound sepharose beads. UBL modification was detected by western blot using the indicated antibodies. Representative blots of three independent experiments are shown. MW, molecular weight marker. **d** Experimental outline for the identification of di-glycylated lysines using a di-glycyl-specific immunoselection coupled to mass spectrometry analysis

5 h post infection (p.i.). MOI, multiplicity of infection. This illustration was created with biorender. **e** Sequence alignment of UB/UBLs. The di-glycyl motif is highlighted in yellow, the N-terminal amino acid, which determines trypsin cleavage, is highlighted in gray. **f–h** Conservation analysis of the identified UBL acceptor lysines in PB2, PB1, and PA. PB2-K113, -K627; PB1-K278, -K586 are not depicted. PB2, PB1, and PA sequences from human, swine, and avian IAV isolates were downloaded from NCBI (PB1 and PB2: 09/15/2017; PA: 10/01/2017) and analyzed for sequence identity. Relative frequency of lysines at the respective position in PB2 (**f**), PB1 (**g**), and PA (**h**) is depicted. In case of a lysine frequency below 75%, the most abundant AA is shown (R = Arginine). Source data are provided as a Source data file.

determined polymerase activity (Fig. 3a), intracellular localization and generation of infectious recombinant viruses (Fig. 3b). Both mutations lead to the loss of the UB/UBL acceptor function but either neutralize (A) or retain (R) the positive charge of the naturally encoded lysine.

Overall, introduction of A was tolerated at 22 positions without affecting the polymerase activity (Supplementary Fig. 2a–c). At 16 sites, A mutations significantly changed the polymerase activity, however, this effect was reverted by the R mutation, which suggests that it was the positive charge of the lysine rather than the presence of the UB/UBL that determined its role for polymerase activity (Supplementary Fig. 2d–f, Supplementary Table 1). For the 17 remaining positions, A and R substitution significantly up- or down-regulated the polymerase activity, which corroborated their function as UB/UBL acceptor sites and suggested a regulatory role of the modification on

the activity of the viral polymerase. This was further supported by the fact that these lysines are located within binding regions to NP, the other polymerase subunits and pol-II as well as the NLS and RNA binding motifs, which are essential domains for polymerase activity (Fig. 3c–e). Notably, previously reported NEDD8 and UB acceptor sites in PB2, such as K699[34] and K482[31], were also among the identified regulatory lysines. Among all three subunits, we identified only two positions that indicated an antiviral effect of the UB/UBL modifications. This included PB2-K157 located in the mRNA exit tunnel and PB1-K635 in the thumb domain. Both mutations demonstrated increased polymerase activity and resulted in the generation of stable recombinant viruses (Fig. 3c, e). Substitution of PA-K22 with A or R resulted in diverging effects on polymerase activity. While PA-K22R reduced polymerase activity, the introduction of A led to a strong

**Table 1 | Modified lysines in the IAV polymerase**

| Domain | K-ε-GG position | Rep. of 5 | Exposed in monomer | | Exposed in dimer | | Rec. virus (titers [PFU/ml]) | | Ref. |
|---|---|---|---|---|---|---|---|---|---|
| | | | vRNA | cRNA | Symmetric | Asymmetric | Ala | Arg | |
| **PB2** | | | | | | | | | |
| N-Terminal domain | 32 | 2 | – | – | – | – | | | [85] |
| | 41 | 4 | – | – | + | – | | | |
| | 61 | 4 | + | + | + | + | | | |
| | 80 | 4 | + | + | + | + | | | |
| | 116 | 4 | + | + | + | + | | | |
| | 121 | 4 | – | – | – | Rep – \| Enc + | | | |
| | 126 | 4 | + | + | + | + | | | |
| | 157 | 2 | + | + | – | – | + ($2.0 \times 10^7$) | + ($1.8 \times 10^7$) | |
| | 187 | 4 | + | + | + | – | | | |
| | 189 | 4 | + | + | – | – | | | |
| | 197 | 2 | + | + | + | + | | | |
| Mid CapB Linker | 312 | 2 | + | – | – | – | | | |
| | 353 | 2 | + | + | + | + | | | [44] |
| | 482 | 4 | + | + | + | Rep+\| Enc – | LT ($5.8 \times 10^4$) | SP ($1.7 \times 10^6$) | [31] |
| 627 Domain | 586 | 2 | + | + | + | + | | | [86] |
| | 617 | 2 | – | + | – | – | – | – | |
| | 670 | 2 | – | + | + | Rep – \| Enc + | | | [87] |
| NLD | 699 | 3 | + | – | + | + | + ($1.0 \times 10^7$) | – | [34] |
| | 718 | 2 | + | + | + | – | SP ($4.8 \times 10^6$) | + | [46] |
| | 752 | 4 | + | + | + | + | | | [31,46,47] |
| **PA** | | | | | | | | | |
| Endonuclease | 22 | 4 | + | + | + | + | + ($1.4 \times 10^8$) | SP ($1.4 \times 10^7$) | [57] |
| | 29 | 5 | + | + | + | + | | | |
| | 102 | 2 | + | + | + | + | | | [48,50] |
| | 104 | 1 | + | + | + | + | | | [48,50] |
| | 113[a] | 2 | + | + | + | + | | | [48] |
| | 134 | 4 | – | – | – | + | – | – | [48,49,88,89] |
| | 158 | 1 | + | – | + | – | | | [49] |
| Linker | 213 | 3 | + | + | + | + | | | [48] |
| | 245 | 3 | + | + | + | + | | | |
| | 251 | 4 | + | + | + | + | | | |
| C-terminal domain | 262 | 4 | + | + | + | + | | | |
| | 281 | 2 | – | – | – | – | | | [42,87,90,91] |
| | 309 | 2 | + | + | – | Rep +\| Enc – | | | |
| | 339 | 2 | + | + | + | Rep +\| Enc – | | | [9,48] |
| | 353 | 1 | + | + | + | + | | | [11,12] |
| | 391 | 1 | + | + | + | + | | | |
| | 536 | 4 | + | + | + | Rep +\| Enc – | R/LT ($7.3 \times 10^3$) | R ($3.6 \times 10^7$) | [53] |
| | 605 | 4 | + | + | + | Rep +\| Enc – | SP ($2.2 \times 10^7$) | + ($7.8 \times 10^6$) | |
| | 609 | 2 | + | + | + | + | | | |
| | 615 | 2 | + | + | + | Rep +\| Enc – | – | + ($1.2 \times 10^7$) | [92] |
| | 626 | 3 | + | + | + | + | | | [93] |
| | 635 | 5 | + | + | + | + | – | SP/LT ($3.0 \times 10^4$) | [51] |
| | 643 | 3 | – | – | – | – | | | [91,93] |
| **PB1** | | | | | | | | | |
| Core | 121 | 3 | – | – | – | – | | | |
| | 229 | 4 | – | – | – | – | – | – | [55,87,94–96] |
| ß-hairpin | 347 | 2 | – | – | – | – | – | – | |
| | 353 | 2 | – | – | + | + | | | [55] |
| | 360 | 2 | + | + | + | + | + ($3.6 \times 10^7$) | LT ($1.1 \times 10^5$) | |
| | 379 | 2 | + | + | + | + | | | |

**Table 1 (continued)**

| Domain | K-ε-GG position | Rep. of 5 | Exposed in monomer | | Exposed in dimer | | Rec. virus (titers [PFU/ml]) | | Ref. |
|---|---|---|---|---|---|---|---|---|---|
| | | | vRNA | cRNA | Symmetric | Asymmetric | Ala | Arg | |
| Palm | 471 | 1 | – | – | – | – | – | SP ($1.1 \times 10^7$) | [97] |
| | 480 | 3 | + | + | + | + | | | [42,55,98] |
| Thumb | 578 | 2 | + | + | + | + | + ($2.2 \times 10^7$) | R ($6.4 \times 10^3$) | |
| | 586 | 1 | + | + | + | + | | | |
| | 635 | 1 | – | – | + | – | + ($9.4 \times 10^7$) | + ($1.2 \times 10^8$) | |
| | 669 | 2 | – | – | – | + | | | [99] |
| C-Ext | 736 | 2 | + | + | + | + | | | |
| | 737 | 4 | + | + | + | + | | | |

List of the positions of all K-ε-GG bound lysines in the PB2, PA, and PB1 subunits detected in 5 independent replicates (Rep.) at 5 h p. i. together with their location in functional domains. The number of replicates in which the site was detected is indicated (Rep. of 5). Surface exposure of the identified sites was assessed using 3D structural models of a WSN-adapted IAV polymerase monomer bound to either vRNA or cRNA as well as the 3D structures of the symmetric dimer of the H3N2 polymerase (PDB: 6QNW) and the ANP32A-stabilized asymmetric dimer of the influenza C virus polymerase. Surface exposure is indicated with (+), concealed location is indicated with (–). In the asymmetric dimer, surface exposure is also itemized regarding location in the replicating (Rep) or encapsidating (Enc) polymerase. Success or failure to generate recombinant viruses with the plasmids encoding the respective mutations after three independent rescue attempts is encoded by (+) = generation of virus, (–) = no virus after three attempts, (R) = generation of virus but reversion to wild type, (SP) = generation of virus but small plaque phenotype, (LT) = generation of virus but low rescue titer. Rescue titers are provided as [PFU/ml].
Studies that have previously described or functionally characterized the detected lysines are listed in the references (Ref.) column.
Following reanalysis of the MS data additional modified sites in PB2 (K113, K627) and PB1 (K278) were detected.
ᵃOne site in PA (K113) that was identified in previous analyses was excluded due to threshold adjustments.

increase to more than 250%. With regard to its location in a hinge region of the PA endonuclease domain and an interaction site with PB1[57] (Supplementary Fig. 2j, Supplementary Table 1), this could suggest, that UB/UBL-mediated neutralization at PA-K22 affects mRNA transcription through structural alterations. Mutations at eight positions abrogated or reduced the polymerase activity but nevertheless gave rise to viable recombinant viruses (Fig. 3c–e, bottom panels, Table 1). We speculated that this discrepancy was facilitated by residual vRNA replication. To confirm this hypothesis, we determined synthesis of the different viral RNA species with strand-specific qRT-PCRs for mRNA and cRNA[58]. Indeed, all mutations ablated mRNA transcription but retained at least 40% cRNA synthesis activity (Fig. 3f–h). In the case of PB1-K360A cRNA synthesis was even upregulated to more than 250% compared to the wild-type (WT) polymerase (Fig. 3h). This indicates that UB/UBL modifications at these positions are particularly required for mRNA transcription. Residue PA-K536 resides within a putative RNA template binding groove[53,54]. Both the introduction of A and R were not stable but caused immediate reversion to K536 during virus rescue, demonstrating a strong selection pressure for lysine. We speculate that neither a constantly neutral nor a positively charged residue is tolerated at this position and that at some point during viral replication, the positive charge of PA-K536 needs to be neutralized, hypothetically by the conjugation of UB/UBL.

Positions in domains with catalytic activity, such as the endonuclease in PA (K134) and the polymerase catalytic cavity in PB1 (K229, K347, K471) were in general highly sensitive to substitution with A or R leading to loss of polymerase activity and lack of recombinant virus generation despite robust protein expression (Fig. 3d, e, Supplementary Fig. 2h, i). PB2-K617, located in a putative NP binding region[41,59], was likewise sensitive to mutation, pointing towards a pro-viral function of the UB/UBL modification (Fig. 3c).

The mutations PB2-K699R and PA-K615A did not support the generation of recombinant viruses despite increased polymerase activity and WT-like levels of cRNA synthesis activity (Fig. 3c, d, f, g), suggesting a supportive function of the modification for other steps in the viral life cycle, which might be hindered by the loss of the modification. In the case of PB1-K578, A and R mutations both strongly elevated the polymerase activity (Fig. 3e). However, while K578A gave rise to stable recombinant virus, rescue with K578R resulted in infectious plaques in only one out of three experiments (Fig. 3i), indicating

that substitution of K578 with a neutral or constantly positive charged residue differentially affects the viral polymerase. The low stability of K578R was further corroborated by reduced hemagglutination inhibition (HI) titers and low viral titers reaching less than $10^4$ PFU/ml after virus rescue in R1 (Fig. 3i and Supplementary Fig. 2k). Nevertheless, virus titers increased to comparable levels of the WT virus after only one additional passage of the rescue supernatant, indicating either reversion of the mutation or introduction of compensatory mutations. In contrast, PB1-K578A did not show reduced viral and HI titers for up to 5 passages after rescue (Fig. 3j, Supplementary Fig. 2l).

Some mutations also affected the intracellular localization of the polymerase subunits. We observed an increased cytoplasmic localization of PB2-K482R and PB2-K699R (Fig. 3l), which are both known to be involved in nuclear-cytoplasmic shuttling of PB2[46,60]. In addition, PA-K635A resulted in an increased translocation of PA into the nucleus (Fig. 3m, n). In summary, our functional analysis revealed that host-derived site-specific UB/UBL modifications in the polymerase subunits confer both positive and negative effects on the activities of the viral polymerase. Furthermore, our results provide evidence that several modifications distinctly impact viral mRNA transcription or vRNA replication.

## PB1-K578 is an acceptor site for UB and is highly conserved among influenza viruses

The different effects of A/R mutations of K578 on viral rescue prompted us to investigate this UB/UBL acceptor site in more detail. We first elucidated whether K578 is modified by UB. Co-expression of wild-type PB1 and K578A with a UB-expressing plasmid resulted in significantly reduced UB levels (~70%) for PB1-K578A despite 14 other UB sites detected in the MS screen (Fig. 4a), which suggested that PB1-K578 represents a major UB target site in PB1. Treatment with the UB-specific peptidase USP2 removed residual UB from both WT and mutant PB1. To obtain structural insights on how UB modification of PB1-K578 affects the polymerase activity, we analyzed the structural environment within the WSN-adapted 3D models of the IAV WT polymerase. PB1-K578 resides in a surface exposed α-helix within the PB1 thumb domain that is part of a positively charged patch constituted of several positively charged residues in PB1 (R571, R572) and PB2 (R101). In the modeled structure, PB1-K578 is directly oriented towards the negatively charged residue PB2-E72, which resides at the tip of a mostly unstructured and flexible loop in the PB2-N1 domain (AA 61–82),

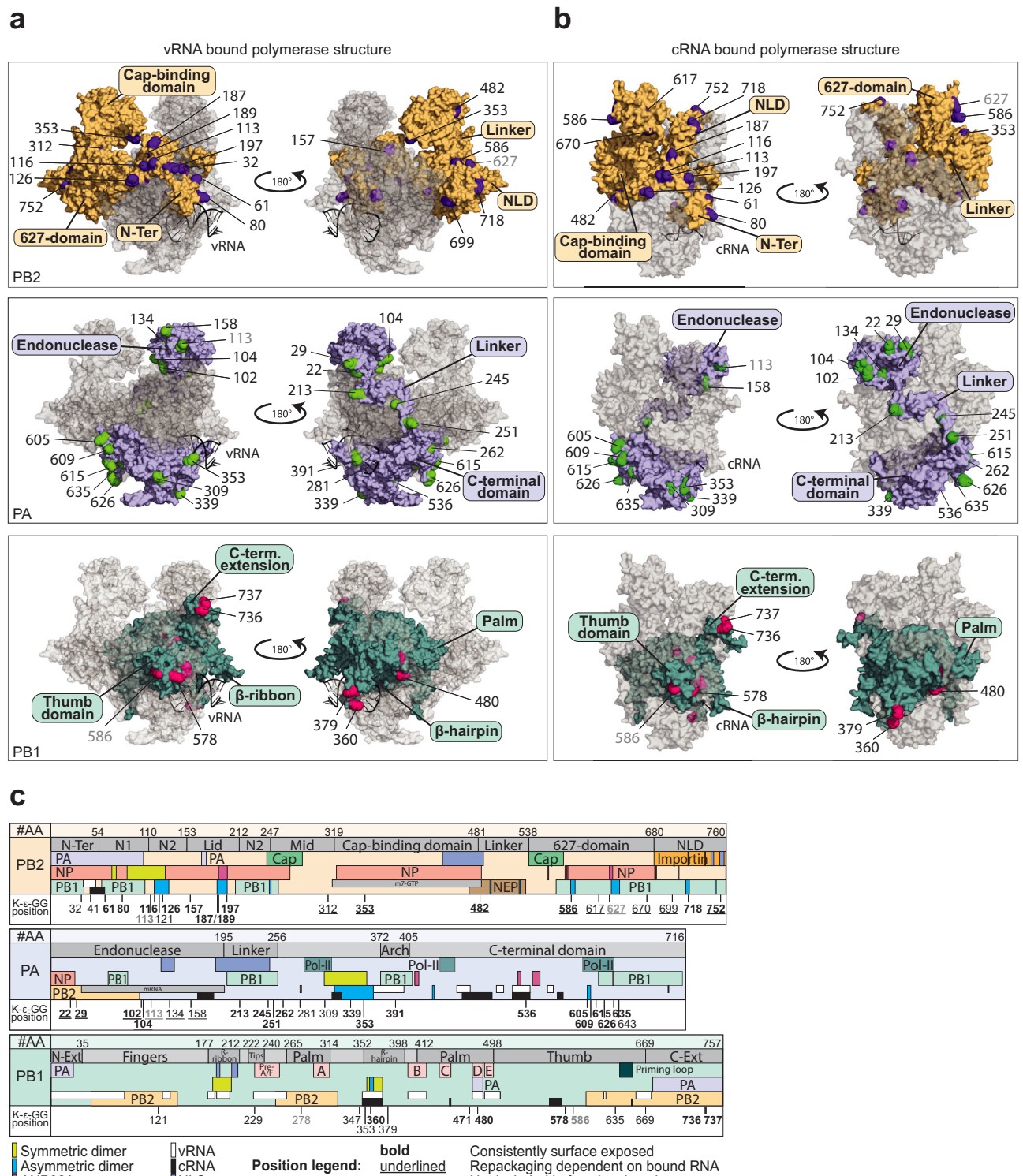

**Fig. 2 | UB-modified lysines reside in functional domains of the IAV polymerase.**
**a**, **b** 3D structural models of a WSN-adapted heterotrimeric IAV polymerase bound to vRNA (adapted from PDB: 4WSB; **a)** or cRNA (adapted from PDB: 5EPI; **b)** created by comparative homology modeling. Positions of lysines with diglycyl remnants are depicted in violet (PB2, yellow, upper panel), green (PA, violet, middle panel), and magenta (PB1, turquoise, lower panel). **c** Linear models of PB2 (upper panel), PA (middle panel), and PB1 (lower panel) depicting the location of the identified di-glycylated lysines (K-ε-GG position) in previously described functional domains.

Boundaries of functional domains (gray boxes) are depicted in the upper lane (#AA). Interaction sites to viral proteins PA (violet), PB1 (light green), PB2 (orange), NP (dark orange), and NEP (brown) and the cellular proteins pol-II (dark green), importin (bright orange), and ANP32A (pink) are included. RNA interaction sites are depicted as follows: mRNA (gray) vRNA (white boxes) or cRNA (black boxes) and the 5′Cap-structures of cellular mRNAs (bright green boxes). Other functional motives include the catalytic domains of the polymerase in PB1 (light pink), NLS (violet), and the priming loop (dark green).

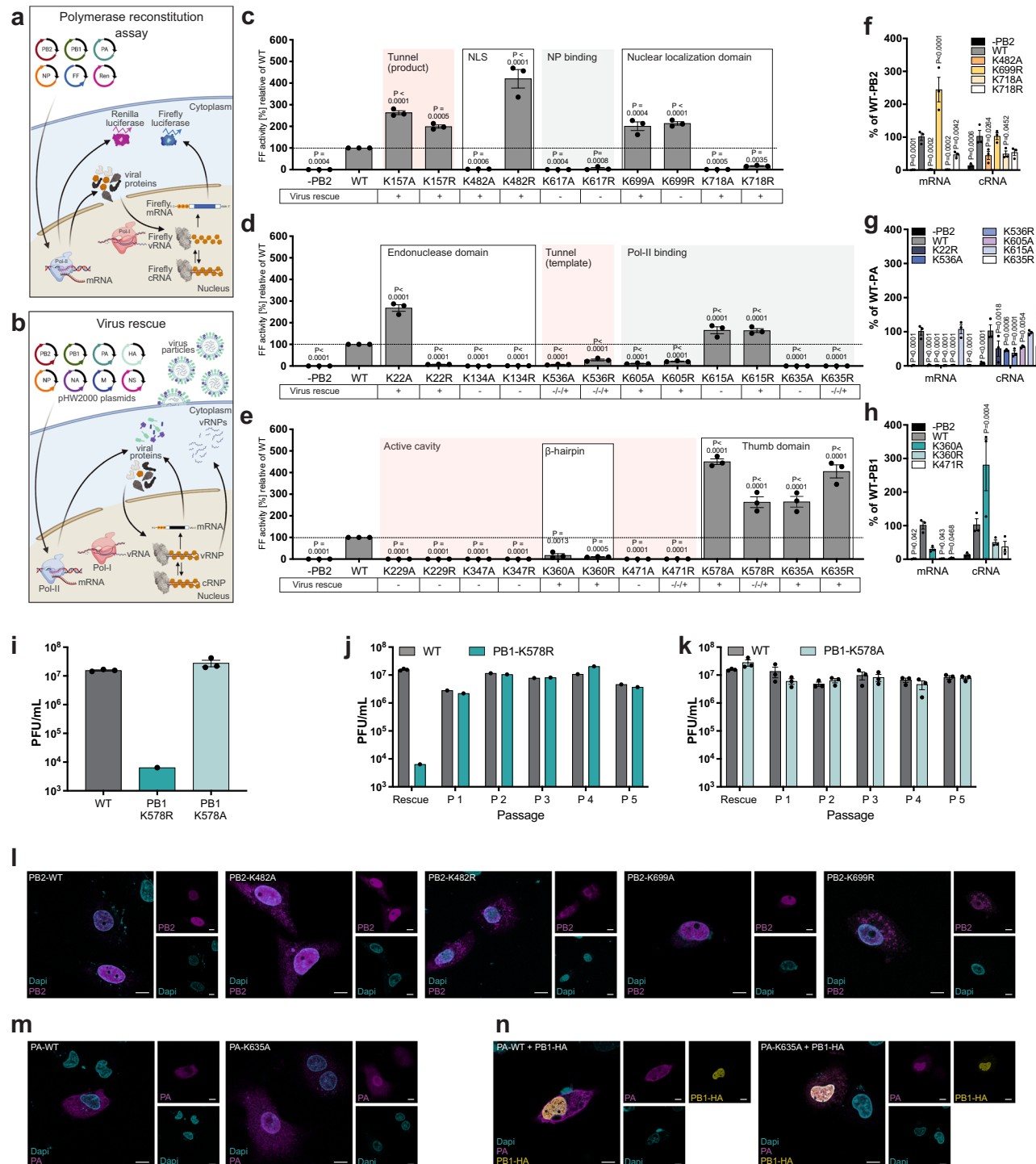

suggesting a direct charge-dependent communication between both residues (Fig. 4b). The PB2 loop was described before to participate in dimerization of the viral polymerase trimer[11,12]. Particularly, residues PB2-T76 and R70 were shown to interact with PA-D347 of the second polymerase (Fig. 4c). In addition, alanine substitution of the triplet PB2-N71-E72-Q73 in the PB2 loop resulted in loss of polymerase activity and dimerization suggesting a crucial role of this loop for viral RNA synthesis[11,12].

The amino acid sequences of the PB1 helix and the PB2 loop are both highly conserved among IAV strains including natural isolates from human, swine and avian hosts (Supplementary Fig. 3a, b). K578, is the predominant amino acid in PB1 sequences derived from human (98.82%), avian (99.26%), and swine (75.18%) isolates (Supplementary

Fig. 3c). Only 0.03% of avian IAV strains contained A578, which was not present in human and swine IAV isolates, suggesting low evolutionary stability in nature. Interestingly, of the analyzed swine PB1 sequences 12.47% encoded for R578 and 12% for S578, a putative acceptor site for phosphorylation. To assess the functional conservation of PB1-K578 we substituted PB1-K578 with A and R in PB1 of IAV strains PR8 (H1N1) and SC35M (H7N7), which resulted in enhanced polymerase activity and resembled the results of the WSN RdRP (Supplementary Fig. 3d). Furthermore, we identified structural resemblance of the PB1-PB2 interface in the polymerases of influenza B and C as well as bat influenza viruses (Supplementary Fig. 3e–g). Taken together, these results reveal that K578 is a highly conserved UB acceptor site in the PB1 subunit of IAV.

**Fig. 3 | Mutational screen of modified lysines reveals distinct effects on mRNA transcription and vRNA replication. a** Schematic of the polymerase reconstitution assay. Cells are co-transfected with plasmids for PB2, PB1, PA, and NP together with a firefly luciferase-encoding vRNA minigenome under control of the pol-I promoter and a Renilla luciferase under control of the pol-II promoter. FF activity correlates with the activity of the viral polymerase. Renilla activity serves as an internal transfection control. **b** Generation of recombinant influenza viruses. Cells are co-transfected with 8 bi-directional pHW2000 plasmids encoding the viral genome segments. Viral mRNA and vRNA is generated from pol-II and pol-I promoters, respectively. Illustrations created with biorender. **c**–**e** Polymerase reconstitution assay with A/R-substitutions of the modified lysines in PB2 (**c**), PA (**d**), and PB1 (**e**). Relative FF activities are presented as the mean percentage activity using the wild type (WT) polymerase (±SEM), *n* = 3 independent biological replicates. *P* values < 0.05 compared to WT from Dunnett's multiple comparison one-way ANOVA test are indicated. Success (+) or failure (−) to generate recombinant viruses

is depicted below. Virus rescue was considered negative if three independent rescue attempts failed. Sites located in functional domains (white), interaction regions with RNA templates (light red), or viral/host proteins (gray) are highlighted. **f**–**h** Quantification of FF mRNA and cRNA levels from the polymerase reconstitution assay for mutations in PB2 (**f**), PA (**g**), or PB1 (**h**) using qRT-PCR using segment and RNA species-specific primers. Values are depicted as mean relative to WT (±SEM), *n* = 3 independent biological replicates. *GAPDH* served as housekeeping control. *P* values < 0.05 compared to WT from Dunnett's multiple comparison two-way ANOVA are indicated. **i**–**k** Viral titers after rescue (**i**) or passaging (**j**, **k**), *n* = 3 independent biological replicates (±SEM). Viral titers were determined 48 h p.i. as plaque forming units (PFU) per ml. **l**–**n** Immunofluorescence of WT or mutated versions of PB2 (**l**), PA individually (**m**), or in combination with HA-tagged PB1 (**n**)[74]. 24 h post transfection A549 cells were fixed and analyzed using the indicated antibodies. Cell nuclei were visualized using DAPI. Representative cells from one experiment are shown. Scale bar = 10 μm.

## Mutation of PB1-K578 remodels the PB1-PB2 interface by charge-dependent rearrangement of a flexible N-terminal PB2 loop

To predict how UB-mediated charge neutralization or a constant positive charge at K578 would affect the structural integrity and dynamic of the putative PB1-PB2 interaction site, we performed extended molecular dynamic (MD) simulations of 100 ns using WSN-adapted 3D models of the IAV WT polymerase in comparison to models harboring K578A and K578R (*n* = 4 for each condition). Simulations were restricted to an area of 45 Å around the PB1-PB2 interface. The analysis resembled that four amino acids are involved in ionic interactions with PB2-E72, including PB1-R571, PB1-R572, PB1-K578, and PB2-R101, of which K578 represents the only UB acceptor site allowing for charge neutralization. Comparing the lifetime of ionic interactions with one of these residues revealed that charge deletion in K578A eliminates PB2-E72 exerted ionic interactions (2.8 ± 2.8%, SEM, *n* = 4, *p* value 0.0076, Fig. 4d), whereas for K578R simulations, lifetime of PB2-E72 ionic interactions (48.3 ± 14.9%) was not significantly different (*p*-value 0.7289) to lifetime evaluated for WT simulations (42.2 ± 7.2%). Because alteration of interactions can result in reorientation of the involved residues, we analyzed the change of residue position of PB2-E72 in dependence of either WT, K578A, or K578R over the complete simulation time by calculating the root mean square deviation (RMSD) every 0.1 ns (4000 values for each condition). However, while this did not reveal a significant shift in a certain mean position of PB2-E72 (WT: 3.160 ± 0.21, K578R: 3.410 ± 0.39, K578A: 4.196 ± 0.58), RMSD distribution analyses demonstrated clear differences for both mutants, indicating that K578A strongly increased the degree of mobility for PB2-E72, since the residue is able to occupy positions with RMSD values of more than 6 Å compared to the initial structure (Fig. 4e). In comparison to the WT simulations, RMSD distribution of PB2-E72 for K578R showed a second peak, indicating the presence of an additional favored position of E72. Interestingly, we observed that mutation K578R also heavily affected the RMSD distribution of the adjacent residue PB2-Q73 (Fig. 4f). While RMSD values for WT and K578A are similar distributed showing the highest peak around 4.5 (K578A) to 5 Å (WT), K578R resulted in a clear shift of the highest peak to 3.25 Å, indicating a different spatial stabilization of Q73 in a conformation different from the conformation observed for WT.

Loss of the charge-dependent interactions (K578A) also significantly increased the flexibility (root mean square fluctuation, RMSF) of residues located at a proximal region of the PB2 loop, spanning AA 63–68, while residues at a distal region of AA 78–82 where not affected (Fig. 4g, Supplementary Table 2). This uneven increment of flexibility could influence the spatial structure of the complete loop. Contrary to K578A, K578R showed no significant RMSF alterations in comparison to WT.

To analyze whether these predictions are supported in the polymerase reconstitution assay, we mutated additional residues in the positive interaction surface (PB2-R101, R571, R572) as well as the PB2

loop (PB2-E72 and Q73). This demonstrated that charge neutralization of PB2-R101A as well as PB2-E72 increased the polymerase activity similar to K578A (Supplementary Fig. 3h, i). In contrast, PB1-K571A resulted in a mild reduction while PB1-K572A and PB2-Q73A abrogated the polymerase activity, suggesting additional interactions with other residues that affect the polymerase activity by a different mechanism of action.

Taken together, the results of the dynamic structure simulations support the previous observation that PB2-E72 is coordinated by a positively charged interaction surface involving PB1-K578 and that mutations PB1-K578A/R both affect the PB1-PB2 interface in a different manner. Neutralization of PB1-K578 interrupted the interaction to PB2-E72 (PB1-K578A) leading to increased flexibility in the proximal part of the PB2 loop. In contrast, K578R preserved the positively charged surface and did not impact the flexibility of the loop compared to WT but instead conferred pronounced repositioning and stabilization of the neighboring residue PB2-Q73. Presumably, this stable position of the PB2 loop is not compatible with the dynamic structural rearrangements in the viral polymerase that are required for viral replication. These results suggest that UB modification of K578 promotes viral replication by a charge-dependent structural rearrangement of the PB1-PB2 interface that may also affect formation of the symmetric dimer.

## PB1-K578R aborts vRNA replication during (multicycle) infection

In contrast to results from the polymerase reconstitution assay, which did not show a negative effect of K578R on polymerase activity, the inefficient viral rescues suggested that a constant positive charge at position PB1-578 is not tolerated during viral replication and poses a strong selection pressure for reversion or alternative mutations. Indeed, deep sequencing of purified infectious and non-infectious virus particles from the supernatants after the rescue experiments revealed a high prevalence of adenines at nucleotide positions 1732–1734 in R1, demonstrating a high frequency of reversion from the mutated 578R to the natural K (Fig. 5a). In contrast, in non-infectious particles from replicates R2 and R3 the distribution of nucleotides at these positions was more heterogeneous, with 0 and 1% adenines at the first position in R2 and R3, respectively, suggesting absence or very low reversion to 578 K. Of note, R2 and R3 contained fractions encoding for 578 A or Q, but nevertheless failed to produce infectious virus also after additional passaging. Compensatory mutations in the polymerase or other genes were not observed.

Analyses of the abundance of the individual vRNA segments within virus particles from rescue supernatants of PB1-K578R, PB1-K578A and the WT virus demonstrated that all eight vRNA segments were present in viral particles from the rescues giving rise to viable virus (Fig. 5b). In contrast, the non-infectious viral particles from R2 and R3 of the PB1-K578R rescue contained incomplete viral genomes,

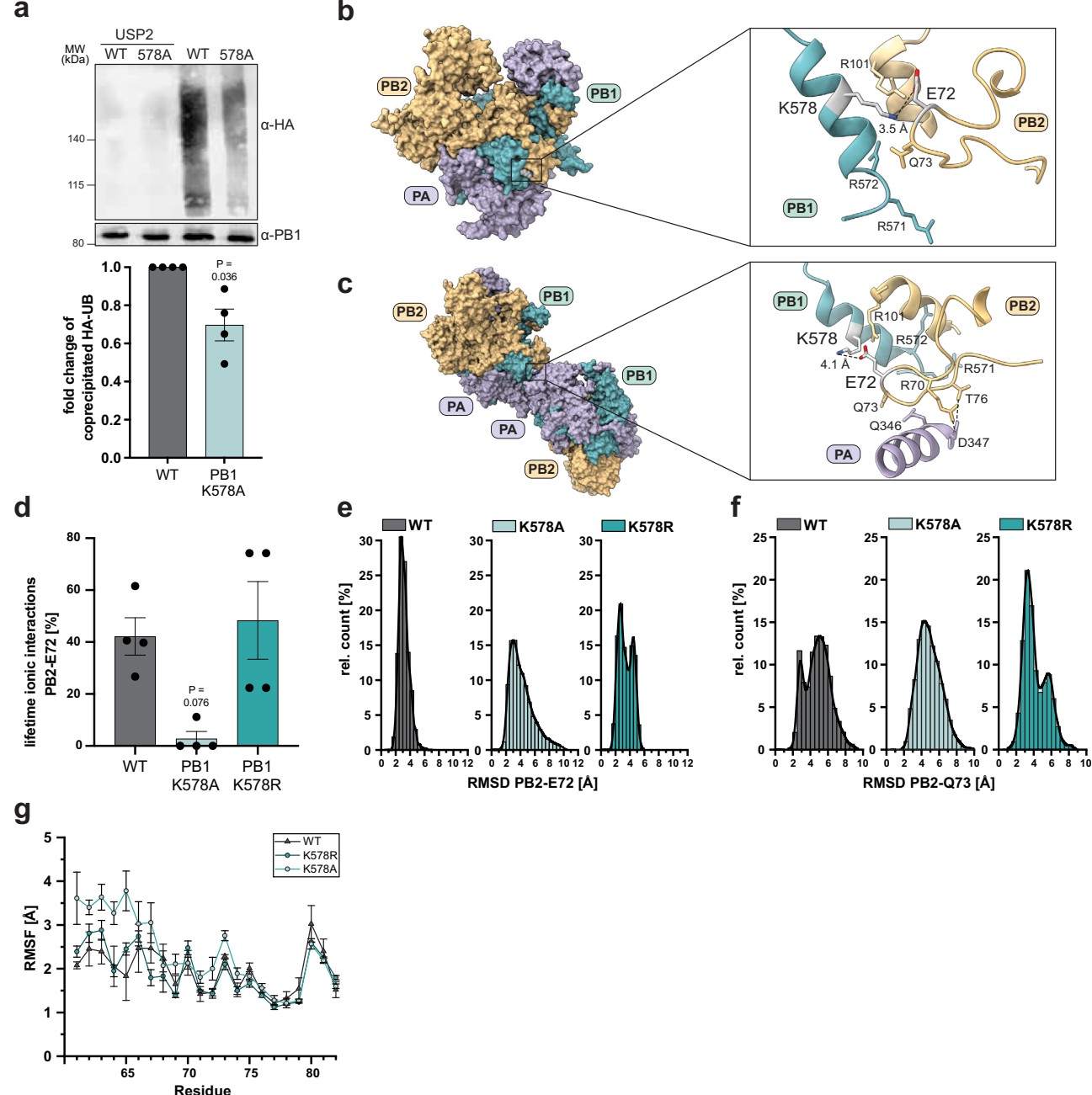

**Fig. 4 | PB1-K578 is ubiquitinated and interacts with a loop in the PB2 N-terminus. a** A549 cells were co-transfected with strep-tagged PB1 or PB1-K578A and HA-tagged UB. UB-modified PB1 subunits were strep-purified using denaturing conditions and analyzed by western blot. For de-ubiquitination, bound PB1 constructs were treated with USP2. Co-precipitated UB-HA levels were quantified and presented as the mean fold change of WT (±SEM), $n = 4$ independent biological replicates. *P* values compared to WT from Welch's corrected two-tailed t-test are indicated. **b, c** Illustration of PB1-K578 in the 3D structures of the WSN polymerase (vRNA-bound conformation; **b**) and the 3D structure of the symmetric dimer of the H3N2 polymerase (PDB: 6QNW; **c**), showing the K578 containing PB1-helix (cyan), the flexible loop in the PB2 N-terminus (yellow) harboring PB2-E72 and the helix in the PA C-terminus that participates in dimer formation. Distances between PB1-

K578 and PB2-E72 are indicated in Angstrom (Å). Amino acids involved in the dimer interface are indicated. Created with ChimeraX. **d** Mean lifetime of ionic interactions with PB2-E72 for 100 ns simulations, $n = 4$ independent biological replicates for each condition (±SEM). *P* values < 0.05 compared to WT from Welch's corrected two-tailed t-test are indicated. **e, f** RMSD distribution of PB2-E72 (**e**) and PB2-Q73 (**f**) for WT, K578R and K578A simulations. Each 100 ns simulation generated 1000 RMSD values leading to a total number of 4000 values for each condition. Equal bin sizes with 0.5 Å steps were used for each histogram. **g** RMSF values of PB2 residues (±SEM). Significance of mean differences was analyzed by Welch corrected two-sided t-test, $n = 4$ independent experiments. *P* values are summarized in Supplementary Table 2. Source data are provided as a Source data file.

implying that defective vRNA replication accounts for the generation of non-infectious virus particles in these replicates.

To determine the step of vRNA replication that was affected by K578R, we next compared the ability of reconstituted WT and mutant polymerase complexes to stabilize and transcribe a cRNA

template provided by viral infection. Cells were first transfected with plasmids expressing PB2, inactive PB1 (PB1a), PA, and NP for 24 h followed by infection with WT virus in presence of cycloheximide (CHX) to inhibit translation of viral proteins. Both, K578A and K578R did not differ in cRNA stabilizing activity compared to the

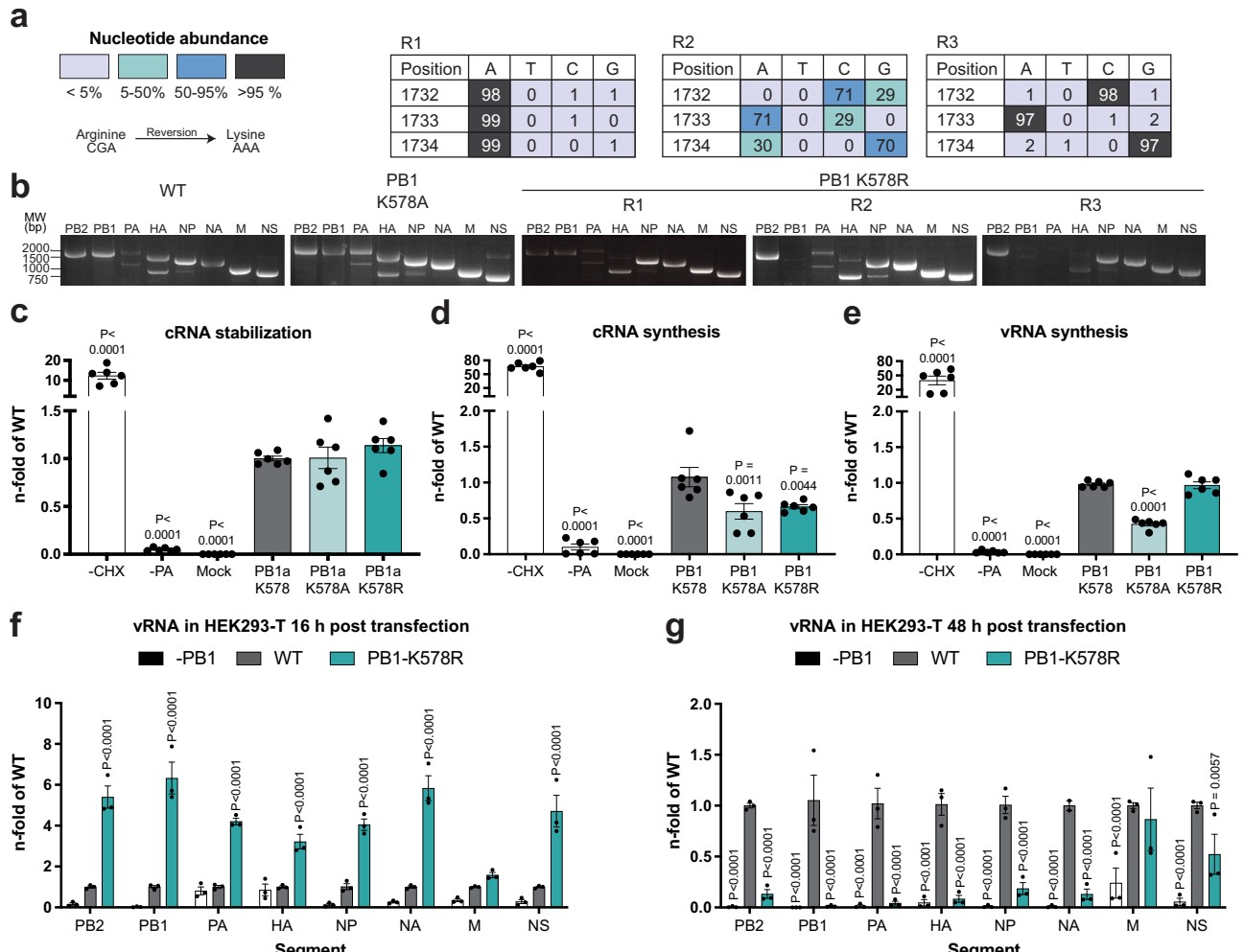

**Fig. 5 | Mutation of PB1-K578 affects cRNP stabilization and vRNA synthesis.**
**a**, **b** Detection of packaged vRNA segments isolated from released virus particles in the supernatants of the rescue experiments at 48 h p.t. vRNA segments were amplified with segment-specific primers, separated on 2% agarose gel (**b**). Gels from all three replicates of K578R and representative gels of three replicates (R) with consistent results are shown for WT and PB1-K578A. vRNA was analyzed using deep sequencing. Percentage of nucleotides encoded at position 1732–34 encoding for K578 (**a**). **c**–**e** Analysis of cRNA stabilization, cRNA and vRNA synthesis. HEK293-T cells were transfected with plasmids expressing NP, PB2, PA, inactive PB1 [PB1a] (D445A/D446A[62,100]; **c**) or active PB1 (**d**, **e**). At 24 h p.t., cells were infected with WT WSN virus (MOI: 5) and cultured in infection medium with cycloheximide (CHX). At

6 h p.i. cRNA (**c**, **d**) or vRNA levels (**e**) of the NA segment were assessed using strand-specific RT-PCR and are shown as mean n-fold of WT (±SEM), $n = 6$ independent biological replicates. $P$ values < 0.05 compared to WT from Dunnett's multiple comparison one-way ANOVA-test are indicated. **f**, **g** Accumulation of vRNA segments in HEK293-T cells co-transfected with pHW2000 plasmids at the indicated time points. vRNA levels were quantified using segment-specific primers for RT-PCR and presented as the mean n-fold of WT (±SEM), $n = 3$ independent biological replicates. Cellular *GAPDH* mRNA levels were used as housekeeping control. $P$ values < 0.05 compared to WT from Šidák's corrected multiple comparison two-way ANOVA test are indicated. Source data are provided as a Source data file.

WT polymerase (Fig. 5c). However, pre-transfection with plasmids for the assembly of an active polymerase complex resulted in significantly reduced cRNA levels for both K578A and K578R (Fig. 5d). In addition, we observed that the neutral K578A polymerase produced significantly lower levels of vRNA compared to the WT as well as K578R polymerase, suggesting less efficient cRNA to vRNA synthesis (Fig. 5e). These results reveal that both mutations hamper a step proceeding cRNA stabilization and that neither a constant positive nor a neutral charged residue at this position supports the synthesis of similar high cRNA levels compared to the WT polymerase, suggesting that UB-mediated charge neutralization of PB1-K578 is required to occur with distinct timing in the infected cell for optimal vRNA levels.

Since these results did not unravel the specific defect of the K578R mutant to generate viable virus, we next compared the amounts of segment-specific vRNAs during virus rescue directly from the transfected HEK293-T cells at 16 h and 48 h post transfection (p.t.) of the 8

rescue plasmids. This demonstrated significantly higher levels of 7 vRNA segments (excluding the HA encoding segment) for the K578R mutant compared to the WT at 16 h p.t. (Fig. 5f). However, this advantage was lost at 48 h p.t. when vRNA levels of all segments were strongly reduced compared to the WT (Fig. 5g), indicating that the defect in vRNA replication of the PB1-K578R harboring polymerase involves the incoming vRNP and is therefore detrimental in infected rather than transfected cells.

## The charge at PB1-K587 modulates NP binding and polymerase dimerization
Accumulating evidence suggests that vRNA replication relies on polymerase dimerization as well as the interaction with free NP to encapsidate newly produced cRNAs and vRNAs[9,11,61–64]. Based on the previous results, we speculated that the alteration in vRNA synthesis by mutation of PB1-K578 could underlie changes in NP binding or polymerase dimerization.

Co-immunoprecipitation of strep-tagged NP with the viral polymerase subunits showed that PB2 and to a minor extent also PB1 directly interacted with NP, while there was no direct interaction with PA (Supplementary Fig. 4a). We further investigated whether the PB1-NP interaction was affected by mutations K578A or K578R but did not find a difference, suggesting absence of a direct interaction at this residue (Supplementary Fig. 4b). In the next step we co-expressed the trimeric polymerase using wild type PB1 or PB1 harboring mutations at PB1–578 with strep-tagged NP and used the PA subunit as a readout for interaction. This demonstrated that PB1-K578A significantly destabilized the interaction between the polymerase and NP (Fig. 6a). In the presence of viral RNA, however, binding of PB1-K578A to NP was not reduced (Fig. 6b), while PB1-K578R did not affect NP interaction in either setting. Collectively, these results suggest, that UB-mediated charge neutralization of PB1-K578 reduces NP binding to the RNA-free trimeric IAV polymerase but not to the vRNP-bound polymerase.

Finally, we analyzed whether the dimerization between two heterologous polymerase trimers was affected by PB1-K578A/R mutations and used the previously described dimerization mutant PB2-71–73 as a positive control for reduced formation of the symmetric dimer[12]. Co-expression of strep-PA and HA-PA together with the other subunits and subsequent strep-pull down resulted in reduced levels of co-precipitated HA-PA for PB1-K578A but not for PB1-K578R, suggesting that UB-mediated charge neutralization also reduces formation of the symmetric polymerase dimer (Fig. 6c). Taken together, these results indicate that UB modification of PB1-K578 modulates assembly of the symmetric polymerase dimer and binding to NP.

## Discussion

Post translational modifications provide a crucial regulatory layer to control the structure and function of viral proteins as well as their intracellular location and interaction to other proteins. Thereby, PTMs can serve as dynamic molecular switches to coordinate the different steps of viral replication.

Here, we unraveled the first comprehensive landscape of site-specific ubiquitin modification in the IAV viral polymerase during infection. The combination of di-glycyl-targeted peptide enrichment and MS analysis resulted in the identification of 59 UBL-modified lysines across all three subunits of the polymerase, the majority of which have not been described nor functionally characterized before. Most lysines are highly conserved in IAV isolates from humans, birds, and pigs and reside within functionally and structurally important regions, emphasizing their putative impact on viral replication. Although di-glycyl remnants derive from different UBL modifiers, we demonstrate that ubiquitin is the most abundantly detected modification in the viral polymerase, which is consistent with earlier reports[27,34]. Several other groups reported site-specific UB/UBL modifications in the IAV polymerase and NP with effects on viral replication. However, these effects were commonly attributed to proteasomal degradation or destabilization of the modified viral proteins, while specific effects on the polymerase functions were not reported[22,28,34]. Our work provides a new perspective on UB/UBL modifications of the IAV polymerase and demonstrates that such modifications confer site-specific effects on the activity of the viral polymerase and the generation of recombinant viruses through non-proteolytic mechanisms, which substantiates that IAV exploits the cellular UB/UBL system to optimize viral replication. We describe no less than 17 UB/UBL acceptor sites in the viral polymerase, which positively or negatively affect the activity of the viral polymerase on the level of intracellular localization, mRNA transcription, vRNA replication, and generation of recombinant viruses upon substitution with A and R. Interestingly, we found that several mutations resulted in the abrogation of mRNA transcription, however, with minor effects on cRNA synthesis. This provides strong evidence for a broad spectrum of functional roles of site-specific ubiquitination in the temporal coordination of both

processes and emphasizes that the regulatory impact of host-derived and site-specific UB modifications on the viral polymerase activity exceeds our current understanding and thus requires further investigation.

In addition to the description of new ubiquitination sites within the IAV polymerase and their functional characterization, we provide compelling evidence that the UB acceptor site PB1-K578 plays a crucial role for mRNA transcription and vRNA replication. The importance of K578 was revealed by mutational substitution with A and R, which both abrogate UB-modification but resemble UB-mediated charge neutralization or the constant positive charge of a non-modified lysine, respectively. According to the current model for mRNA transcription[5], we speculate that the increased polymerase activity of the PB1-K578A/R mutants observed in the reconstitution assays (Fig. 3e) derives from an altered repositioning of the PB1 thumb domain and the PB2-N1 domain, which alleviates blockage of the template exit channel by PB2 (AA 80–90) in the pre-initiation complex of the transcriptase and thereby promotes mRNA elongation.

Dynamic molecular structure modeling predicted that PB1-K578A negatively affects the integrity of a positively charged interaction surface constituted by PB1-R571, R572, K578, and PB2-101, leading to the spatial reorientation of the entire N-terminal PB2 loop by weakening the charge-dependent interactions to residue PB2-E72 and subsequently influencing spatial positioning of PB2-Q73. Consequently, we found that PB1-K578A had a reduced affinity to NP in the context of the RNA-free polymerase and demonstrated reduced formation of the symmetric dimer. Importantly, this was accompanied by significantly reduced cRNA and vRNA synthesis activity, while cRNA stabilization was not affected. Nevertheless, PB1-K578A supported rescue of a stable recombinant virus. In combination with the defect of the PB1-K578R mutant to rescue a stable virus but instead enforcing the rapid reversion to the natural lysine, as well as the pronounced defect in vRNA replication, our data emphasize that both, the non-modified and the UB-modified forms of K578 are required for viral replication.

The dynamic structure simulations indicate that K578R stabilizes the PB2 loop differently, leading to an altered favored positioning profile of residue Q73. Because PB1-K578R demonstrated WT-like polymerase dimerization and cRNA to vRNA synthesis in the transfection-infection experimental setting (Figs. 5e, 6c) we speculate, that this position of the loop represents the favorable position for assembly of the symmetric dimer. Despite the WT-like behavior in several assays, K578R is detrimental for viral replication and does only allow the generation of viable recombinant viruses when rapid reversion to the natural K is achieved. Based on these results and the currently discussed models of vRNA replication, we speculate that the PB1-K578R encounters at least two problems during infection caused by the absence of UB-mediated charge neutralization (Fig. 6d). First, the incoming vRNP produces high levels of viral mRNAs leading to expression of new trimeric polymerases harboring PB1-K578R as well as free NP. Due to the constant positive charge of PB1-K578R, the free trimeric polymerase has a high affinity to interact with NP, which reduces both, the pools of free trimeric polymerase needed for formation of the different dimers as well as NP that is required for cRNP and vRNP assembly. Secondly, we assume that PB1-K578R promotes premature assembly of the symmetric polymerase dimer possibly already between the incoming vRNP-associated polymerase and newly expressed PB1-K578R harboring polymerase, which leads to abortion of cRNA synthesis. Both mechanisms lead to an imbalanced viral RNA synthesis and will pose a strong pressure for reversion to the modifiable lysine. In reverse conclusion, we propose that ubiquitination of K578 is required to decrease NP binding to newly synthesized trimeric polymerases to retain the free polymerase pool and to achieve correct positioning of the PB2 loop that prevents the premature assembly of the symmetric dimer. The mechanism underlying decreased NP binding to the trimeric polymerase but not the vRNA-associated NP by

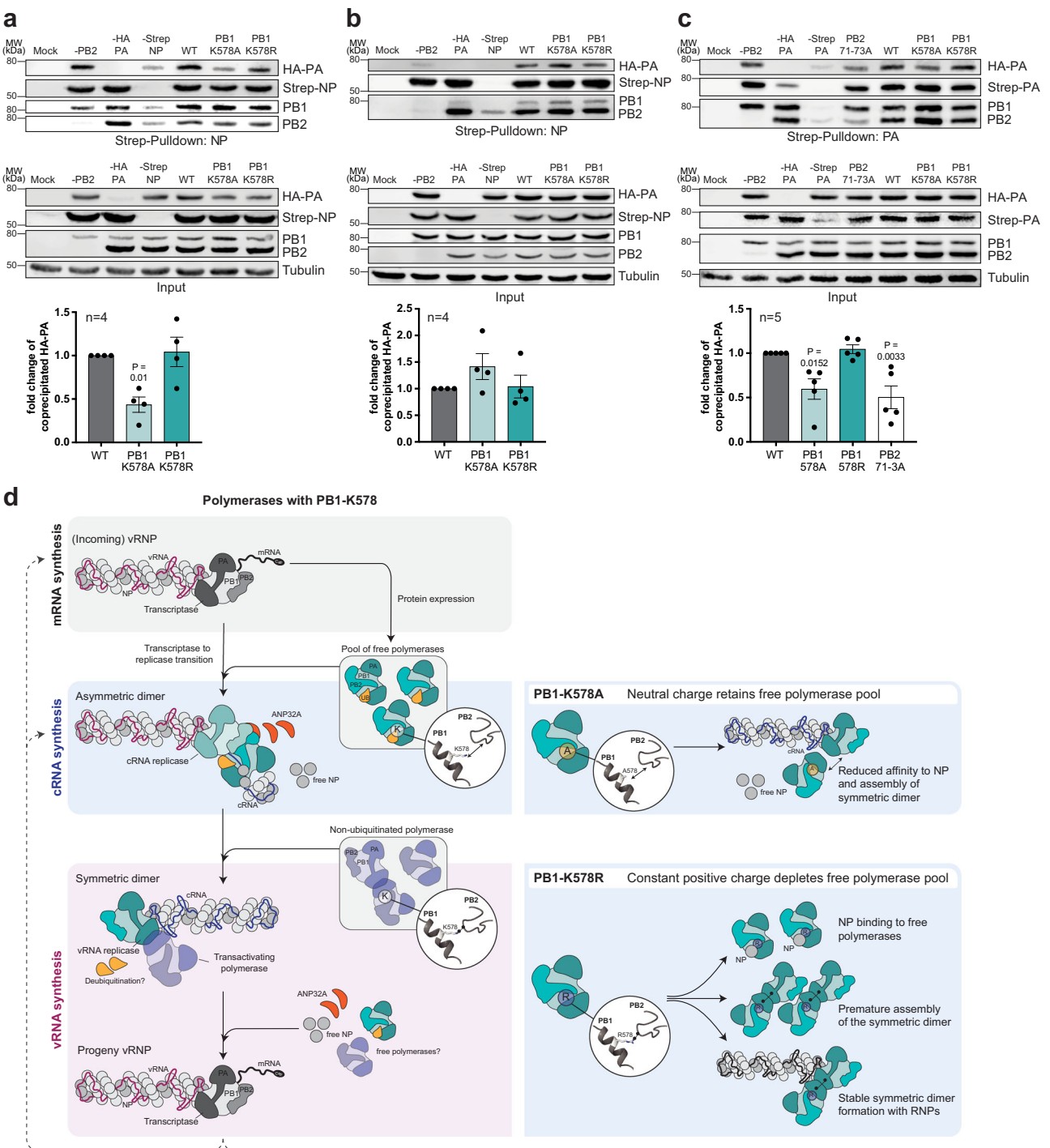

mutation K578A or K578 ubiquitination remains open to speculations. Whether this is also mediated by a charge-dependent mechanism that affects the positioning of the PB2 loop or rather involves shielding of the NP binding site by the UB moiety cannot be concluded from our results. Moreover, it remains elusive whether our results reflect reduced binding of the first NP in the vRNP, or the terminating NP and if NP binding and dimer formation are competitive events due to the overlapping binding sites.

The responsible E3 ligase for PB1-K578 ubiquitination is not known. A previous report demonstrated that among the more than 600 E3 Ubiquitin ligases UBR5, STUB1/CHIP, HUWE1, and DDB1 interact with the viral replication machinery in infected cells[14]. Several groups have previously used comparative interactomics of the human

ubiquitin proteasome system with viral proteins, genome-wide RNAi library screens or mass spectrometry-based interaction screens[22,30,65] to identify candidate E3 ligases that facilitate ubiquitination of viral proteins. The results of our study can now be combined with such technologies to link cellular E3 ligases to site-specific modifications in the viral polymerase and uncover new therapeutic targets. In conclusion, in addition to generating the first comprehensive ubiquitin landscape of the viral polymerase, our results provide a novel and unexpected role of host-derived ubiquitination of residue PB1-K578 in the PB1 thumb domain for the coordination of NP binding, polymerase dimerization and vRNA replication (Fig. 6d). These novel findings advance our understanding of the enigmatic binding site of NP to the polymerase trimer and unravel the important role of the flexible

**Fig. 6 | Mutation of PB1-K578 affects the binding of free polymerase to NP and polymerase dimerization. a, b** Detection of NP binding to trimeric polymerase by co-affinity precipitation. Cells were transfected with PB1, PB2, HA-tagged PA, strep-tagged NP (**a**) and a firefly-encoding vRNA reporter (**b**) followed by strep-purification 24 h p.t. Co-precipitated proteins were detected by western blot. **c** Polymerase dimerization assessed by co-affinity precipitation. Cells were transfected with PB1, PB2 and both HA- and strep-tagged PA. 24 h p.t. the polymerase components bound to strep-tagged PA were purified. Co-precipitated proteins were detected by western blot. (Input Strep-PA signals derive from a blot analyzed in parallel). Quantification of the interaction is shown below the panels. Levels of HA-tagged PA were normalized to strep-tagged NP (**a**, **b**) or strep-tagged PA (**c**) and depicted below as the mean n-fold of WT (±SEM). The number of independent biological replicates (*n*) is provided in the figure. *P* values < 0.05 compared to WT from Dunnett's multiple comparison one-way ANOVA are indicated. Source data are provided as a Source data file. **d** Model depicting the biological function of PB1-K578 ubiquitination during vRNA replication. Incoming vRNPs perform viral mRNA

transcription for viral protein expression including the viral polymerase proteins and NP. A subpopulation of newly synthesized trimeric polymerase complexes acquires ubiquitination at PB1-K578 (cyan polymerase complexes with UB in yellow) which disrupts the interaction between PB1-K578 and the loop in the PB2-N1 domain, prevents early formation of the symmetric dimer and reduces the affinity to NP. The PB1-K578 ubiquitinated polymerase interacts with the vRNP-bound polymerase to form the ANP32A-stabilized asymmetric dimer for cRNA synthesis and encapsidation. For vRNA synthesis from the cRNA template, a non-ubiquitinated polymerase (violet polymerase complexes) interacts with the cRNP-associated polymerase under formation of the symmetric dimer. Ultimately, progeny vRNPs may perform secondary mRNA transcription, further rounds of vRNA replication or get exported out of the nucleus for progeny virion assembly. Substitution of the UB-acceptor site PB1-K578 with both alanine and arginine affects vRNA replication (right panel). While PB1-K578A retains the pool of free polymerases, PB1-K578 aborts progression of vRNA replication due to a depletion of free polymerases.

N-terminal PB2 loop for the formation of the symmetric polymerase dimer. Moreover, our study emphasizes the remarkable potential of UB-modifications to fine-tune protein structures and functions way beyond the mechanism of proteolytic degradation.

## Methods

### Cells

Madin-Darby Canine Kidney cells type II (MDCK-II), human embryonic kidney 293T cells (HEK293-T), and human alveolar epithelial cells (A549) were maintained in Dulbecco's modified Eagle medium (DMEM), supplemented with 10% fetal calf serum (FCS; Capricorn) and 1% Penicillin/Streptomycin (P/S; Sigma). All cells were cultured at 37 °C and 5% $CO_2$. MDCK-II, HEK293-T, and A549 cells were provided by Martin Schwemmle (Freiburg, Germany).

### Plasmids

For generation of recombinant influenza A viruses, the eight reverse genetics bi-directional pHW-2000 plasmids encoding vRNA and viral proteins derived from the A/WSN/1933 (H1N1) (WSN) virus were used[66]. Mutations in pHW-2000 PB2, PB1, and PA plasmids were introduced by site-directed mutagenesis (SDM). To create overexpressing plasmids encoding for mutated PB2, PB1, and PA of WSN, A/PR/8/1934 (PR8), and A/SC35M/1980 (SC35M), the respective pHW-2000 plasmids bearing the respective mutations were used as templates and subjected to assembly PCR to introduce restriction sites for NotI and XhoI restriction enzymes (NEB). The resulted PCR fragments were cloned into pCAGGs plasmids. pCAGGs vectors that encode Strep-tagged polymerase subunit constructs of WSN-derived wild type PB2, PB1 and PA were created likewise. The pCAGGs plasmids encoding C-terminally HA-tagged PB1 and PA fusion proteins were a kind gift from Martin Schwemmle (Freiburg, Germany) and described previously[67]. The pPolI-Firefly reporter plasmid encodes a negative-sensed *luciferase* gene flanked by the 5' and 3' non-coding regions of the NS segment. Likewise, the pPolI-Firefly-UP-Promoter encodes a positive-sensed *luciferase* gene and has been described before[68]. Both were used for minigenome assays together with the pTK-Renilla plasmid. pRK5-HA-Ubiquitin-WT was a gift from Ted Dawson (Addgene plasmid #17608) and pcDNA3-HA-NEDD8 was a gift from Edward Yeh (Addgene plasmid #18711). The pCMV2b-RGS-His-ISG15 WT plasmid was a gift from Gerrit Praefcke (Langen, Germany).

### Expression of viral proteins

HEK293-T cells were transfected with pCAGGs plasmids encoding for PA, PB2, PB1 (WT or mutant), 24 h p.t. the cells were lysed using ice-cold radio immunoprecipitation assay buffer (RIPA; 150 mM NaCl, 0.1% (w/v) SDS, 0.5% (w/v) sodium deoxycholate, 25 mM TRIS (pH 7.5), 1% (v/v) Triton X-100, protease inhibitor cocktail). Clarified cell lysates were adjusted to equal amounts using Pierce™ BCA assay (Thermo

Fisher), mixed with Laemmli buffer (4X), incubated at 95 °C for 5 min, and subjected to SDS-PAGE and western blot. Proteins were detected using the primary antibodies rabbit anti-PB1 (GTX125923, Genetex; 1:2000 in blocking buffer), rabbit anti-PA (GTX125932, Genetex; 1:1000 in blocking buffer), and mouse anti-Tubulin (clone DM1A, Sigma-Aldrich; 1:1000 in blocking buffer) and the secondary antibody anti-Rabbit IgG-800CW (LI-COR; 1:10,000 in blocking buffer), and anti-Mouse IgG-680RD (LI-COR; 1:10,000 in blocking buffer). Uncropped blots are provided in a source data file.

### Detection of post-translational modification by affinity precipitation

For detection of ubiquitination, A549 cells were cotransfected with pCAGGs plasmids expressing nStrep-PB2-, nStrep-PB1-, or nStrep-PA fusion proteins with HA-ubiquitin using X-tremeGENE HP DNA Transfection Reagent (Sigma). Due to the reduced expression levels of plasmids that encode HA-tagged NEDD8 or His-tagged ISG15 in A549 cells, the modification of the polymerase subunits with these UBLs was performed in HEK293-T cells by co-transfection of the respective plasmids with Lipofectamin 2000 (Invitrogen). A549 cells were processed 48 h post transfection (p. t.) and HEK293-T cells 24 h post transfection as previously described[69]. Briefly, cells were treated with lysis buffer (2% SDS, 150 mM NaCl, 2 mM EDTA, 1% Triton X-100, Protease inhibitor cocktail), proteins were denatured at 95 °C for 10 min, sonicated for 30 s, renatured in a dilution buffer (10 mM TRIS HCl (pH 8.0), 150 mM NaCl, 2 mM EDTA, 1% Triton X-100) for 1 h at 4 °C and then clarified by centrifugation. Clarified lysates were adjusted to equal amounts using Pierce BCA assay (Thermo Fisher) and precipitated using Strep-Tactin sepharose beads (IBA Lifescience) at 4 °C for 2 h. Beads were washed thrice using a stringent washing buffer (10 mM TRIS-HCl (pH 8.0), 1 M NaCl, 1 mM EDTA, 1% Igepal CA-630). In case of deubiquitinase treatment, the beads were washed twice with DUB reaction buffer (1X), as provided in UbiCREST Kit[70], and incubated with USP2 at 37 °C for 30 min according to the manufacturer description. Precipitated proteins were eluted using 1X Laemmli buffer (0.25 M TRIS (pH 6.8), 40% glycerol, 8% SDS, 10% β-mercaptoethanol, 0.01% bromophenol blue) by incubation at 95 °C for 5 min. The precipitated samples were subjected to SDS-PAGE and western blot. Covalently bound modification by HA-ubiquitin, HA-NEDD8, or His-ISG15 of each polymerase subunit was detected using the primary antibodies rat anti-HA-Tag (clone 3F10, Roche; 1:500 in blocking buffer), rabbit anti-NEDD8 (clone Y297, Abcam; 1:1000 in blocking buffer) or mouse anti-His-Tag (clone HIS.H8, Thermo Fisher; 1:1000 in blocking buffer) and secondary antibodies anti-Rat IgG-HRP (Cell signaling Technologies; 1:3000 in blocking buffer), anti-Rabbit IgG - HRP (Cell signaling Technologies; 1:3000 in blocking buffer) and anti-Mouse IgG - HRP (Cell signaling Technologies; 1:3000 in blocking buffer). Polymerase subunits were detected using the primary antibodies rabbit

anti-PB2 (GTX125926, Genetex; 1:1000 in blocking buffer), rabbit anti-PB1 (GTX125923, Genetex; 1:2000 in blocking buffer), rabbit anti-PA (GTX125932, Genetex; 1:1000 in blocking buffer) and the secondary antibody anti-Rabbit IgG-680RD (LI-COR; 1:1000 in blocking buffer). Uncropped blots are provided in a source data file.

## Mass spectrometry analyses of K-ε-GG-enriched peptides
Infection was carried out by incubating $10^8$ A549 cells with IAV WSN in PBS supplemented with 0.7% BSA, 1% $CaCl_2$, and 1% $MgCl_2$ at a multiplicity of infection (MOI) of 20 for 30 min at 4 °C and subsequently for additional 30 min at 37 °C. For the detection of the ubiquitin remnant motif (K-ε-GG) on the IAV polymerase subunits, the PTMScan®-Kit (Cell Signaling Technology) was applied according to manufacturer's protocol. Briefly, 5 h post infection (p.i.) cells were lysed using Urea lysis buffer (20 mM HEPES (pH 8), 9 M urea, 1 mM sodium orthovanadate, 2.5 mM sodium pyrophosphate, 1 mM β-glycerophosphate), followed by sonication. Clarified lysates were reduced and alkylated using dithiothreitol (DTT) and chloroacetamide, respectively, and diluted 3-fold with 20 mM HEPES (pH 8.0). Lysates were trypsin digested by incubation with 1% TPCK-Trypsin (m/v in 1 mM HCl) overnight at 25 °C, desalted using Sep-Pak® C18 cartridges and immunoaffinity purification (IAP) using the anti-(K-ε-GG) motif antibody of the kit. Final purification was performed on Stage tips as described before[71].

Peptides enriched for the ubiquitin remnant motif (K-ε-GG) were dissolved in either 0.5% acetic acid or 0.1% formic acid prior to LC-MS/MS analysis using a LTQ Velos Orbitrap ($n = 1$) or Q Exactive HF mass spectrometer ($n = 3$; Thermo Scientific), respectively. Each MS instrument was online coupled to an EASY nLC nanoflow HPLC system via a Nanospray Flex ion source (Thermo Scientific), harboring in-house packed fused silica capillary column (length 15 cm; ID 75 µm; ReproSil-Pur C18-AQ, 3 µm). Following loading, bound peptides were eluted using a linear 240 min gradient from 3–40% B (80% ACN, 0.5% acetic acid) followed by a short gradient from 35–98% B in 10 min at a flow rate of 300 or 250 nl/min. After washing at 98% B the column was re-equilibrated at starting conditions. The Orbitrap Velos Pro mass spectrometer was operated in the positive ion mode, switching in a data-dependent fashion between survey scans in the orbitrap (mass range $m/z = 300–1650$; resolution R = 60,000; target value = 1e6; lockmass set to 445.120025) and collision induced fragmentation and MS/MS acquisition in the LTQ part. MS/MS spectra of the 15 most intense ion peaks detected in the MS1 scans were recorded. Conditions for the analysis using the Q Exactive HF mass spectrometer were kept similar (Top17 DDA; Full MS R = 60,000; AGC target = 3e6; Maximum IT = 100 ms; scan range 300–1750 $m/z$; ddMS2 R = 15,000; max IT = 50 ms; NCE = 27 V). Dynamic exclusion was enabled on both MS systems.

Raw MS data were processed using MaxQuant (v. 2.0.1.0) with the built-in Andromeda search engine. Tandem mass spectra were searched against the Influenza A virus (strain A/Wilson-Smith/1933 H1N1; UP000000834.fasta; version from 10/2016) as well as against the human UniprotKB database (UP000005640_9606.fasta; version from 04/2019) concatenated with reversed sequence versions of all entries and also containing common lab contaminants. Carbamido-methylation on cysteine residues was set as fixed modification for the search in the database, while oxidation at methionine and acetylation of the protein N-termini were set as variable modifications. For the identification of ubiquitination sites, modification of lysine residues with the remnant diglycyl motif was allowed as additional variable modification. Trypsin was defined as the digesting enzyme, allowing a maximum of two missed cleavages and requiring a minimum length of 7 amino acids. The maximum allowed mass deviation was 20 ppm for MS and 0.5 Da for MS/MS scans. Common lab contaminants and proteins containing reverse sequences that were derived from the decoy database were filtered out from the dataset prior to any further analysis. Protein groups were regarded as being unequivocally identified with a false discovery rate (FDR) of 1% for both the peptide and protein identifications. Ubiquitination sites were accepted when they were identified with a localization probability of >0.75 (class I sites). PX partial: All mass spectrometry data have been deposited to the ProteomeXchange Consortium (http://proteomecentral.proteomexchange.org) via the PRIDE partner repository with the dataset identifier PXD030816, DOI: 10.6019/PXD030816[72].

## Sequence analysis
PB1, PB2, and PA sequences from isolates of human, swine and avian IAV origin were downloaded from NCBI. 2004 sequences for IAV PB2 and 1630 sequences for IAV PB1 were downloaded on 09/15/2017. 2016 sequences for IAV PA were downloaded on 10/01/2017. The cd-hit was used to assign clusters of 99% sequence identity. Sequences of each protein were aligned using MUSCLE (multiple sequence comparison by Log-Expectation).

## Polymerase reconstitution assay
HEK293-T cells were cotransfected with 50 ng of pCAGGs plasmids expressing PA, PB2, PB1 (either WT or mutant) together with 200 ng of pCAGGs-NP plasmids and 25 ng of both pTK-Renilla plasmids and the vRNA expressing pPolI-Firefly or the cRNA expressing pPolI-Firefly-Up-Promoter plasmids using Lipofectamin™ 2000. 24 h post transfection cells were lysed and relative polymerase activity was examined using the Dual-Luciferase Reporter Assay (Promega) and reporter gene expression was determined by measuring luminescence using a MicroLumat Plus LB96V luminometer (Berthold Technologies).

## Generation of recombinant viruses
For generating recombinant viruses co-cultivated HEK293-T cells and MDCK-II cells were co-transfected with the eight pHW-2000 plasmids derived from WSN (WT or mutant) using Lipofectamin™ 2000 (Invitrogen). After 6 h of incubation in Gibco™ Opti-MEM™ I Reduced Serum Medium (Thermo Fisher) at 37 °C the supernatant was removed and cells were incubated with infection medium (DMEM supplemented with 0.7% BSA, 1% $CaCl_2/MgCl_2$, and TPCK-Trypsin) for 48 h at 37 °C. The virus rescue supernatant was plaque purified on MDCK-II cells by plaque assays using Oxoid™ purified agar (Thermo Fisher). Then, MDCK-II cells were infected with single plaques diluted in infection PBS and incubated in infection medium for 48 h at 37 °C. Final virus stocks were obtained by passaging this supernatant once on MDCK-II cells with a defined MOI of 0.001. To control for the respective mutation, MDCK-II cells were infected with an MOI of 2. After 6 h, cells were lysed and RNA was extracted using the RNeasy Plus Mini Kit (Qiagen). The respective gene segment was amplified using OneStep RT-PCR Kit (Qiagen) and sequenced by Sanger sequencing (Eurofins Genomics).

## Viral genome sequencing
Following isolation of the viral RNA from virus supernatant using QIAamp Viral RNA Mini Kit (Qiagen), each vRNA segment was reverse transcribed using the SuperScript™ III One-Step RT PCR System with Platinum™ *Taq* DNA polymerase (Invitrogen). In the following, approximately 1 ng of the pooled PCR products was used as template and introduced into library preparation with the Nextera XT DNA Sample Preparation Kit (Illumina) and paired-end sequenced with the 2 × 250 bp MiSeq Reagent Kit v2 (Illumina) with an average insert size of 300 bp on a MiSeq instrument (Illumina). The protocols for library preparation and sequencing were conducted as recommended by the manufacturer (Illumina). After automatic demultiplexing on the MiSeq instrument, the resulting fastq files were mapped onto the Influenza A virus strain A/WSN/1933(H1N1) reference sequence (GenBank accession numbers CY034132-CY034139) using the BWA mapping algorithm implemented in the Ridom SeqSphere⁺ software (v7) with default parameters. Subsequently, variant positions (detection via the

software with default parameters) were checked manually and the variant read frequencies and percentage of each nucleotide at the respective position were determined.

## Accumulation of viral segments during virus rescue

To assess the accumulation of viral segments during virus rescue with limited virus propagation HEK293-T cells were cotransfected with the eight pHW-2000 plasmid system derived from WSN using Lipofectamin™ 2000. 16 h and 48 h post transfection RNA was isolated using TRIzol (Thermo Fisher). Residual DNA was digested using recombinant DNase I (Roche). The RNA was reverse transcribed with oligo(dT) or Uni12 primers using RevertAid H Minus Reverse Transcriptase (Thermo Fisher Scientific). Briefly, primers and 500 ng RNA were heated at 70 °C for 5 min, immediately chilled on ice for 1 min, and then heated at 37 °C for 2 min. Then, a mixture of First Strand Buffer (5x, Invitrogen), dNTPs, and RevertAid H Minus Reverse Transcriptase (Thermo Fisher Scientific) was added and incubated at 37 °C for 10 min, followed by an incubation at 42 °C for 60 min. Reverse transcription was terminated by incubation at 85 °C for 5 min. CT values were measured in duplicates using a LightCycler® 480 II (Roche) and the LightCycler® 480 Software (v1.5.1.62), normalized to the expression data of the housekeeping gene *glyceraldehyde 3-phosphate dehydrogenase (GADPH)* and analyzed using the $2-\Delta\Delta CT$ method[73].

## Quantification of viral RNA by strand-specific RT-PCR

HEK293-T cells were transfected with 250 ng of pCAGGs plasmids expressing WT or mutant PB2, PB1, and PA, 1000 ng of NP and 100 ng of a pol-I driven minigenome firefly reporter plasmid expressing vRNA, using Lipofectamin™ 2000. 16 h p.t. RNA was isolated using TRIzol (Thermo Fisher) and dissolved in RNase-free water. Residual DNA was digested using recombinant DNase I (Roche). For reverse transcription a mixture of tagged primers directed to either m-, c-, or vRNA of the firefly minigenome (designed according to Kawakami et al.[58]), oligo-dT primers, and 250 ng RNA were heated at 65 °C for 10 min, immediately chilled on ice for 5 min and then heated at 60 °C for 5 min. Then, a mixture of First Strand Buffer (5x, Invitrogen), 0.1 M dithiothreitol, dNTPs, and Maxima H Minus Reverse transcriptase (Thermo Scientific) was added and incubated at 60 °C for 60 min. Reverse transcription was terminated by incubation at 85 °C for 5 min. RT-PCR was carried out using Brilliant III SYBR Green (Agilent) and primers directed to the respective tags. CT values were measured in duplicates using a Light-Cycler® 480 II (Roche) and the LightCycler® 480 Software (v1.5.1.62), normalized to the expression data of the housekeeping GADPH and analyzed using the $2-\Delta\Delta CT$ method. Primer sequences are included within Supplementary Data 1.

## cRNP stabilization assay

HEK293-T cells were cotransfected with pCAGGs plasmids that express for PB2, PA, and NP together with pCAGGs plasmids for either WT or inactive PB1 (PB1a:D445A/D446A) harboring the indicated mutations using Lipofectamin™ 2000. After maintaining the cells on Gibco™ Opti-MEM™ (Thermo Fisher) for 6 h, the medium was exchanged to DMEM (supplemented with 10% FBS and 1% P/S). 24 h p.t. the cells were infected with WSN WT virus at an MOI of 5 and maintained on infection medium (supplemented with 100 μg/ml cycloheximide (CHX)). For controls, PA was substituted with an empty vector, cells were mock infected or CHX was omitted. 6 h p.i., RNA was isolated using TRIzol (Thermo Fisher). For reverse transcription the protocol of strand-specific qPCR (as described above) was applied using tagged oligo-dT primers as well as primers for c- and vRNA of the NA segment. CT values were measured in duplicates using a LightCycler® 480 II (Roche) and the LightCycler® 480 Software (v1.5.1.62), normalized to the expression data of the housekeeping gene *GADPH* and analyzed using the $2-\Delta\Delta CT$ method. Primer sequences are included within Supplementary Data 1.

## Immunofluorescence analysis

A549 cells were seeded on glass coverslips and transfected with pCAGGs plasmids expressing either WT or mutants of PB2, PB1 (in combination with nHA-PA), and PA (in combination with nHA-PB1)[74] using XtremeGene™ (Roche) 24 h p.t., cells were fixed with 3.7% formaldehyde, permeabilized with 0.1% Triton X-100 and blocked with 3% BSA in PBS (blocking buffer). For immunostaining coverslips were incubated with the primary antibodies rabbit anti-PB2 (GTX125926; Genetex; 1:3000 in blocking buffer), rabbit anti-PB1 (GTX125923, Genetex; 1:3000 in blocking buffer), rabbit anti-PA (GTX125932, Genetex; 1:3000 in blocking buffer), rat anti-HA-Tag (clone 3F10, Roche; 1:500 in blocking buffer), overnight at 4 °C and with secondary antibodies anti-rabbit Alexa Fluor 488 (Invitrogen; 1:2000 in blocking buffer) or anti-rabbit Alexa Fluor 568 (Invitrogen; 1:2000 in blocking buffer) and anti-rat Alexa Fluor 488 (Invitrogen; 1:2000 in blocking buffer) for 1 h at room temperature. Cell nuclei were stained with DAPI (Thermo Fisher Scientific) for 20 min at room temperature. Coverslips were mounted using Mounting Medium S3023 (Dako Omnis) and examined using an LSM-800 Airyscan confocal microscope (Carl Zeiss) and ZEN software (v2.6).

## Hemagglutination inhibition assay

To quantify the total amount of virus particles after infection, the volume of the assessed supernatants was adjusted to a final volume of 50 μl based on the PFU titer using infection PBS. The virus samples were serially diluted (1:2) in infection PBS until dilution 1:1024 in a V-bottom-shaped microtiter plate. Then, 50 μl of fresh human erythrocytes (blood type: 0) was added to each well, mixed with the virus dilutions, and incubated for 1 h on 4 °C.

## Affinity precipitation of NP-polymerase complexes

HEK293-T cells were transfected with pCAGGs plasmids encoding for Strep-tagged NP (nStrep-NP) fusion protein, HA-tagged PA (nHA-PA), PB2, PB1 (WT or mutant), and a pol-I driven minigenome firefly (FF) reporter plasmid using Lipofectamin® 2000. For negative controls, nStrep-NP, PB2, and nHA-PA plasmids were substituted with an empty vector. 24 h p.t. the cells were lysed using ice-cold Co-IP lysis buffer (50 mM TRIS (pH 8.0), 150 mM NaCl, 5 mM EDTA,1% (v/v) Igepal CA-630, protease inhibitor cocktail), as described before[75]. Clarified cell lysates were adjusted to equal amounts using Pierce™ BCA assay (Thermo Fisher) and precipitated using Strep-Tactin sepharose beads (IBA Lifescience) for 2 h at 4 °C. Beads were washed 5 times using Co-IP washing buffer (50 mM TRIS (pH 8.0), 500 mM NaCl, 5 mM EDTA, 1% (v/v) Igepal CA-630). Precipitated proteins were eluted in Laemmli buffer (1X) and subjected to SDS-PAGE and western blot. Co-precipitated proteins were detected using the primary antibodies rabbit anti-NP (GTX125989, Genetex; 1:1000 in blocking buffer), rat anti-HA-Tag (clone: 3F10, Roche; 1:5000 in blocking buffer), rabbit anti-PB1 (GTX125923, Genetex; 1:1000 in blocking buffer), rabbit anti-PB2 (GTX125926, Genetex; 1:1000 in blocking buffer) and mouse anti-Tubulin (clone DM1A, Sigma-Aldrich; 1:1000 in blocking buffer) and the secondary antibodies anti-Rabbit IgG-800CW (LI-COR; 1:10,000 in blocking buffer), anti-Rat IgG-HRP (Cell signaling Technologies; 1:3000 in blocking buffer) and anti-Mouse IgG-680RD (LI-COR; 1:10,000 in blocking buffer). Uncropped blots are provided in a source data file.

## Dimerization of the viral polymerase

HEK293-T cells were transfected with pCAGGs plasmids expressing nStrep-PA, nHA-PA, WT, or mutant PB1 and PB2, using Lipofectamin™ 2000 (Invitrogen). For negative controls nStrep-PA, nHA-PA, and PB2 plasmids were substituted with an empty vector. After 24 h cells were lysed in 400 μl of TRIS lysis buffer (50 mM TRIS-HCl (pH 7.6), 200 mM NaCl, 25% glycerol, 0.5% Igepal CA-630, 1 mM DTT, protease inhibitor cocktail) for 30 min at 4 °C, as described before[12]. Clarified cell lysates were adjusted to equal amounts using Pierce™ BCA assay (Thermo

Fisher) and precipitated using Strep-Tactin sepharose beads (IBA Life-science) at 4 °C for 16 h. Beads were washed 5 times with a washing buffer (10 mM TRIS-HCl (pH 7.6), 150 mM NaCl, 10% glycerol, 0.1% Igepal CA-630) and proteins were eluted using 1x Laemmli buffer. Precipitated proteins were detected by western blot using the primary antibodies rat anti-HA-Tag (clone: 3F10, Roche; 1:500 in blocking buffer), mouse anti-Strep-Tag (clone GT661, Sigma-Aldrich), rabbit anti-PB1 (GTX125923, Genetex; 1:1000 in blocking buffer), rabbit anti-PB2 (GTX125926, Genetex; 1:1000 in blocking buffer) and mouse anti-Tubulin (clone DM1A, Sigma-Aldrich; 1:1000 in blocking buffer) and the secondary antibodies, anti-Rat IgG-800CW (LI-COR; 1:10,000 in blocking buffer), anti-Mouse IgG-HRP (Cell signaling Technologies; 1:3000 in blocking buffer), and anti-Rabbit IgG-680RD (LI-COR; 1:10,000 in blocking buffer). Uncropped blots are provided in a source data file.

### Homology modeling of WSN polymerase structures

For both models of IAV WSN polymerase bound to vRNA as well as the IAV WSN polymerase model bound to cRNA comparative homology modeling was performed using MODELLER (v9.19). For the vRNA-bound polymerase model the crystal structure of the heterotrimeric Bat Influenza A polymerase bound to the vRNA promoter (PDB: 4WSB) served as a template, for the cRNA-bound polymerase model the crystal structure of Influenza B virus in complex with 5′ cRNA (PDB: 5EPI) served as a template. Crystal structure sequences were aligned with the polymerase subunit sequences of WSN ((i) A/WSN/1933(H1N1) PA (GenBank: CY034137); (ii) A/WSN/1933(H1N1) PB1 (GenBank: CY034138 and (iii) A/WSN/1933(H1N1) PB2 (GenBank: CY034139) and 100 homology models (including loop models) were created of both conformation states based on these WSN sequences. The RNA was transferred as a rigid body from the respective original structure. A 'slow' optimization protocol and 'slow' molecular dynamics refinement was further applied. The linker regions were modeled using "loop model" without imposing any restraints on its structure. The best models were selected based on a normalized energy score. The models were scored using the normalized DOPE score available in MODELLER. Subsequently, superposition models were prepared by VMD (v1.9.3) and figures were generated using PyMOL (v2.3.0, The PyMOL Molecular Graphics System, Schrodinger LLC) or ChimeraX (v1.3.)[76,77].

### Molecular dynamic simulations

The homology model of the three-dimensional structure of the WSN polymerase complex bound to cRNA was used for further structural analyses. The residue PB1-K578 was mutated to K578A or K578R leading to three independent models for further simulations. Each model was prepared and energy minimized for local simulations by initialization procedure of YASARA structure (v21.6.2 and v21.8.27)[78] using AMBER14[79] force field including structure cleaning and hydrogen network optimization[80,81], generation of a water shell (TIP3P) around the protein model as well as prediction of $pK_A$ values at the chosen pH of 7.4[82]. After short equilibration simulation (~5 ns, AMBER14) of the whole protein, the models were subjected to 100 ns local molecular dynamic simulations. Since previous results clearly identified the PB1-PB2 loop-interface as the crucial region for altered behavior of K578R and K578A, molecular dynamic (MD) simulations were conducted to evaluate the interplay between these two domains. To achieve a simulation time sufficient for interplay analyses, local MD simulations were performed, which was only possible due to the strong separation of single functional domains in the protein complex. Especially the PB1-PB2 loop-interface can be identified as an independent domain of the complex, that encompasses a spatial dimension equal to a 45 Å-sphere. Therefore, atoms in a 45 Å-sphere around residue PB1-K578 including only protein residues were set mobile, while all other atoms were immobilized. Further, the size of the simulation box was limited to 92.95 × 92.95 × 92.95 Å, which is sufficient to include a fully mobile water shell, all mobile protein residues as well as enough immobilized

protein residues to ensure stability and integrity of the protein. On the other hand, this procedure reduces the possibility of extensive MD artefact formation due to the presence of large amounts of immobilized residues. Using this approach, we optimized the needed calculation time and computational resources leading to a simulation time of 100 ns for each replicate, which was not achievable in an all-atoms mobile simulation for the complete protein. Local MD simulations were conducted using the following settings: AMBER15IPQ force field[83], particle-mesh Ewald/Poisson-Boltzmann cutoff 8 Å, periodic simulation cell boundary, long range coulomb forces, 0.9% NaCl, pH = 7.4, water density 0.997 (TIP3P), pressure of 1 atm and a simulation temperature of 298 K. Complete simulation including temperature and pressure settings was controlled by a NPT ensemble and atomic motions were integrated with a multiple timestep of 2 × 1.25 fs for bonded interactions and 2.5 fs for non-bonded interactions as previously described[84]. Local simulations were replicated to $n = 4$ for all conditions (WT, K578R, K578A). Each simulation was documented by simulation snapshots every 0.1 ns leading to a total number of 4000 analyzable snapshots for each condition.

The calculation of the RMSF is performed in three automated steps. First, the mean position ($\bar{P}_{ij}$) of the distinct atom ($i$) is calculated using atom position vector (P) with $j = 3$ components for the x, y, and z coordinates of every single snapshot ($k$) from the 100 ns simulation ($N = 1000$). In the second step, the root mean square fluctuation is calculated by the following equation:

$$\text{RMSF}_i = \sqrt{\sum\nolimits_{j=1}^{3} \left( \frac{1}{N} \sum\nolimits_{k=1}^{N} P_{ikj}^2 - \bar{P}_{ij}^{\,2} \right)} \qquad (1)$$

Both steps are performed for all atoms individually. In the final step, the RMSF for each residue is calculated by the average RMSF of its constituting atoms.

### Data analysis and statistics

Quantitative data were analyzed and plotted using GraphPad PRISM™ (v7.04) and Origin 2022 (v9.9). Immune fluorescence images were analyzed using ImageJ (v1.53c). Intensities of immuno-stained western blots were quantified using Image Studio 5.2.5 (Li-COR Biosciences). Biochemistry experiments were performed at least in triplicate unless otherwise stated, and representative gels are shown. Details of statistical tests and replicates are provided in the figure legends.

### Reporting summary

Further information on research design is available in the Nature Portfolio Reporting Summary linked to this article.

## Data availability

The mass spectrometry proteomics data generated in this study have been deposited to the ProteomeXchange Consortium via the PRIDE partner repository with the dataset identifier PXD030816, https://doi.org/10.6019/PXD030816 (http://proteomecentral.proteomexchange.org/cgi/GetDataset?ID=PXD030816). MD simulations were conducted using the commercially available software YASARA dynamics (v21.6.2 and v21.8.27) and the implemented macros "md_run.mcr" and "md_analyze.mcr". Source code of YASARA macros can be downloaded from www.yasara.org. Raw simulation data encompassing around 40 gigabytes are available from the corresponding author. The study made use of the publicly available datasets: NCBI: PB1, PB2, and PA sequences from isolates of human, swine, and avian IAV origin were downloaded on 09/15/2017 (PB2, PB1) and 10/01/2017 (PA). PDB entries: 4WSB, 5EPI, 6QNW, 7NHA, 4WSA, 5D9A, 6T0N, 6TW1, 6T0S, 6T0V, 6SZU, 6T2C, 6T0U, and 2VQZ. GenBank: accession numbers CY034132, CY034133, CY034134, CY034135, CY034136, CY034137, CY034138, and CY034139. UniprotKB database (UP000000834.fasta version from 10/2016; UP000005640_9606.fasta version from 04/

2019). Biochemical data generated in this study are provided in the Source data file. Source data are provided with this paper.

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

## Acknowledgements

We thank Ted Dawson, Edward Yeh, and Gerrit Praefcke for providing plasmids. We owe our thanks to Max Trauernicht, Lars Schmitz, Julius Kindsgrab, Josephine Friedrich, Hannah Junginger, Nadja Rotte, Ramona Wördemann, and Josua Janowski for support and Sebastian Giese and Kevin Ciminski for helpful comments and discussions. We are grateful for the support by Geoffrey Chase for editing of the manuscript. This work was funded by the German Research Foundation (DFG) (Grant BR5189/1-1 to L.B.; LU 477/23-2 to S.L., SCHW 632/21-1 to M.S.) and the Clinical Research Unit CRU342 (Project P6 to L.B. and S.L.; project P4 to A.M.), Further support was received by the Interdisciplinary Centre for Clinical Research (IZKF) (Grant Bru2/015/19 to L.B.) of the Medical Faculty of the University of Muenster and the Swiss National Science Foundation (SNF) (Grant 310030_184947 to S.A.L.). We acknowledge support from the Open Access Publication Fund of the University of Muenster.

## Author contributions

L.B., F.G., and T.K. designed and conceptualized the study. F.G., T.K., L.H., and M.W. performed experiments and functional assays including plasmid cloning, polymerase reconstitution assays, immune fluorescence analysis, viral rescues, and protein interaction studies. T.K., S.L., and H.D. were responsible for protein purification from infected cells and MS analysis. V.C. and L.H. generated the WSN-adapted structure models of the viral polymerase and F.G. performed further structural analysis. In silico dynamic structure modeling of the mutated viral polymerase and viral genome sequencing was performed by J.A.S., G.S., and A.M., respectively. M.S. and S.L. provided laboratory infrastructure, plasmids, and cell lines and supported writing and editing of the manuscript. L.B. and F.G. wrote the first draft of the manuscript and designed the figures with support from all co-authors.

## Funding

## Competing interests
The authors declare no competing interests.
