## [Peer Review File · Nature Communications]

The ubiquitination landscape of the Influenza A Virus polymeraseReviewers' comments:

Reviewer #1 (Remarks to the Author):

This study applied a mass spec approach to map the ubiquitination landscape of the IAV polymerase complex. In the first half of the manuscript, authors provide an overview of mass spec results, nicely presenting the data in a structural context, and mechanistically characterize identified residues through a mutagenesis follow-up. Novel sites with, when ubiquitinated, proviral and antiviral activity were convincingly identified. The second half of the manuscript focusses on residue K578 in the PB1 thumb domain specifically, proposing ubiquitination at this position as a regulator of polymerase dimerization and vRNA replication.

These are important findings that represent a major advance of the understanding of the regulation of influenza virus polymerase activity. Data are overall of high quality and support the major conclusions. However, *in silico* always delivers, especially when based on homology models. The presentation of homology model predictions and molecular dynamics simulations needs to be toned down, even if the models created are very persuasive.

Other points:

- 1) Please speculate on the underlying mechanism coordinating timing of PB1 ubiquitination at K578.
- 2) Mutagenesis of identified ubiquitination sites to A and R is well justified. However, even a very conservative K to R exchange can affect protein interactions and/or bioactivity independent of ubiquitination status. Consider that the alleged pro- and antiviral functions associated with individual residues could alternatively simply reflect a specific requirement for a lysine residue at that particular position.
- 3) I am wondering whether the extremely long text blocks in the second half of the results section can be broken down into more palatable sections to make the study more accessible to a broader readership.

Reviewer #2 (Remarks to the Author):

General comments

In this manuscript, Günl et al. aim to identify and characterise ubiquitination sites on the influenza A virus RdRp. The authors begin by using an immunoprecipitation-mass spectrometry approach to identify

UB and UBL sites, then follow up with a comprehensive mutational analysis of these sites. The authors use these data to shortlist UB/UBL sites of interest and attempt recombinant virus rescues, eventually focussing on the PB1 K578 amino acid residue. The authors then perform a series of assays to interrogate how K578A and K578R mutations impact RdRp activity and protein-protein interactions, concluding that the K578 residue is important for regulating RdRp dimerization through its ubiquitylation.

The manuscript is clearly written, and most experiments are of high quality. The identification of UB/UBL sites and subsequent mutational analysis is a particular strength of this manuscript. While some of the experiments which investigate the PB1 K578 mutations do produce intriguing results, others appear to suffer from technical issues, and some are mis-interpreted. In addition, the data shown do not support the model presented at the end of the manuscript.

Specific points

Fig 6E: This model suggests that ubiquitylation of PB1 K578 controls viral genome replication by inhibiting formation of the symmetric dimer. However, this model isn't consistent with the dimerization assay shown in Fig. 4L, which demonstrates that the alanine mutation inhibits dimerization while the arginine mutation has no effect. Surely if ubiquitination of K578 inhibited symmetric dimer formation, both of these mutations would show an increase in dimerization as they cannot be ubiquitylated? The fact that the PB2 71-73A mutation has a potent effect in the dimerization assay confirms that this assay is examining the symmetric dimer rather than the asymmetric dimer.

Fig. 4L: This is a key assay for the conclusions of the manuscript; however, the controls do not make sense: Why is Strep-PA not pulled down in the -PB2 control? Why is a significant amount of HA-PA being detected in the pulldown even with no Strep-PA present in the -PB2 control? Why is there a clear Strep-PA band in the input even in the -Strep-PA control?

Fig. 4H-J: This is another key assay for the conclusions of the manuscript, as the authors use these data to suggest that the PB1 K578 mutations affect both cRNA synthesis and vRNA synthesis. However, these results appear to be misinterpreted. cRNA stabilisation assays normally use an active site mutation in the PB1 subunit to prevent vRNA synthesis, and this allows cRNA synthesis to be examined in isolation (Vreede et al. 2004). In the assay performed here there does not appear to be an active site mutation, so cRNA accumulation will be dependent on vRNA synthesis and vice versa. Therefore, this assay cannot distinguish cRNA synthesis from vRNA synthesis, and rather illustrates the overall efficiency of viral genome replication.

Fig. 4K: PB1 and PB2 bands are visible in the pulldown sample, but not in the input. Please show the PB1 and PB2 bands in both.

Fig. 4K, L: As these data illustrate a quantitative difference between wild type and K578A, please either show a graph of the quantification with error bars, or include a statement in the legend indicating how many times the experiments have been repeated with similar results.

Fig. 5A: This is a key experiment, as it shows that the modification to PB1 K578 is UB rather than NEDD8. As with Fig. 4K and 4L, this conclusion is based on a quantitative difference, so please either quantify the ubiquitin signal across multiple replicates or include a statement in the legend indicating how many times the experiment has been repeated with similar results.

Fig. 3C, D: For amino acid residues PB2 K482 and PA K22, the alanine/arginine mutations have opposite effects on RdRp activity. How would this make sense if they are UB/UBL acceptor sites as suggested in lines 157-159?

Fig. 3I-K: Please include an image of the DAPI channel only, as this stain isn't clear in the merged image (such as in Fig. 3K, leftmost panel).

Supp. Fig. 2A-F: Please add labels to each panel of the western blots.

Supp. Fig. 2A-F: This figure has panels spliced together with more than one wild type lane. Please make it clear which wild type sample should be compared with each of the mutant samples.

Supp. Fig. 2 legend: Panels G-I are mis-labelled in the legend.

Supp. Fig. 3B: It is not clear whether this is re-blotting samples from the pulldown experiment shown in Fig. 4K, or if it is a separate experiment. Either way, please include all relevant controls for a pulldown assay, such as those included in 4K.

Supp. Fig. 4H legend: Typo, PB2-E72 is repeated twice.

Table 1: It would be good to include the virus titres which were obtained from successful rescues.

Lines 143-145: From this point on, it is assumed in the text that every amino acid residue identified is modified by ubiquitin. This may not be the case, as the authors can clearly observe NEDDylation of all RdRp subunits in addition to ubiquitylation (Fig. 1A). It should be made clear in the following sections of the manuscript that these could be ubiquitylation or NEDDylation sites.

Lines 177-180: This is overly speculative.

Lines 313-318: The statement that the PB1 K578R mutation promotes the symmetric dimer interaction is not consistent with the dimerization assay in Fig. 4L, which shows that K578R has no effect.

Please remove speculative conclusions and tone down the abstract.

Reviewer #3 (Remarks to the Author):

The authors studied the ubiquitination of polymerase of influenza A virus by the host E3 ligase during infection. They identified that mutations of PB1-K578A and K578R which prevented ubiquitination by E3 increased the polymerase activity, albeit having an adverse effect on cRNP stabilization. The authors used molecular dynamics simulations to explain the effect of the mutations concluding that the charge neutralization of K578 caused by ubiquitination facilitates viral replication by increasing the flexibility of PB2 loop.

The results from the molecular dynamics simulations lack the clarity and statistical analysis to support the conclusions drawn. More specifically, the “mobility” of PB2 loop is not directly analyzed. Several important details of the methodology are missing, compromising the reproducibility of the results.

Below are concerns specific to the section “PB1-K578 controls the spatio-temporal position of the PB2 loop in a charge-dependent manner:

1. “This analysis further indicated that K578R resulted in prolonged and less dynamic interaction with residues PB2-Q73 and R101 compared to K578 (Fig. 6C and D), which likely leads to a stabilization of the PB2 loop and thereby promotes the interaction to the trans-activating polymerase that is required for the formation of the symmetric dimer.”

- to support this statement, a dynamic estimate of the interaction should be presented. (e.g. lifetime of the interactions)

2. "These results suggest that the defect in vRNA replication of the K578R mutant is caused by restricted mobility of the PB2 loop and thereby constitutes the selection pressure for reversion to K578. Presumably, neutralization of K578 by ubiquitination provides a molecular switch that determines the stability of the PB2 loop, which facilitates transition of the transcriptase to the replicase and regulates both steps of vRNA replication (Fig 6E)."

- the mobility of the PB2 loop was not analyzed and the changes in the secondary structure shown in fig 6A do not immediately lead to this conclusion. We suggest demonstrating the difference in fluctuations of the PB2 as a more appropriate quantity.

Below are concerns related to the methodology and presentation of the results:

1. The confidence interval and statistical significance of the measured quantities (secondary structure in fig 6A and quantities S41) is not reported. A comparison of the mean values (with confidence intervals and statistical significance) is desired as well as a larger size/resolution of the panel with a focus/highlight of the region of interest.

2. In Fig 6B-D, It's not clear what is the purpose of presenting the intersection of measured "ionic", "hydrophobic" and other quantities, rather than only the quantities themselves. We suggest either removing them or discussing their significance in the text.

3. "The complex interactions were analyzed in detail using Yasara ligand" - which interactions specifically and in what details? Hydrogen bonds can be formed between the sidechains as well as the backbone of the residue (which would be independent of the mutation). We suggest the authors rewrite this statement or at least specify which moiety is involved in the interaction.

4. The rationale for conducting "local" simulations is not explained.

5. Q73 is not labeled in fig. 6A. The coloring of the labeled residues does not appear to correspond to the legend. How were these colors chosen and why?

6. The following details are missing from the methods section:

a. What was the forcefield used for nucleic acid?

b. Reference to the AMBER forcefield missing

c. What temperature was the simulation conducted at and what was the algorithm used to control it?
Same for pressure

d. What was the solvent model used?

e. What was the integrator in the simulations?

f. How was the protonation of residues determined?

g. How were the adjacent residues to K/A/R578 (R101, E72 and Q73) determined?

h. How was the secondary structure “abundance” calculated?

Point-by-point response to the reviewer remarks

Reviewer #1

This study applied a mass spec approach to map the ubiquitination landscape of the IAV polymerase complex. In the first half of the manuscript, authors provide an overview of mass spec results, nicely presenting the data in a structural context, and mechanistically characterize identified residues through a mutagenesis follow-up. Novel sites with, when ubiquitinated, proviral and antiviral activity were convincingly identified. The second half of the manuscript focusses on residue K578 in the PB1 thumb domain specifically, proposing ubiquitination at this position as a regulator of polymerase dimerization and vRNA replication. These are important findings that represent a major advance of the understanding of the regulation of influenza virus polymerase activity. Data are overall of high quality and support the major conclusions. However, *in silico* always delivers, especially when based on homology models. The presentation of homology model predictions and molecular dynamics simulations needs to be toned down, even if the models created are very persuasive.

Response: We thank the reviewer for this overall very positive comment on our ubiquitination screen and functional analysis of the modified lysines in the viral polymerase as well as the appreciation of the novelty and importance of our results on K578 modification for the general understanding of this complex enzyme. We understand that we have in parts overstated the results of the *in silico* analysis. We have now completely redone the *in silico* analysis to provide statistical solidity of our data and refined insights into the interplay of residues PB1-K578 and PB2-E72. This has resulted in novel insight into how the position of the PB2 loop is coordinated (please see new Fig. 4.) In addition, instead of reporting the *in silico* results at the end of the results part, we repositioned the newly written part on the *in silico* analysis before the further experimental analysis and now clearly state in the text that this analysis is a prediction. We hope that this meets the reviewer's concerns. Please find the newly written part and the corresponding figures on the *in silico* analysis in lines 262-315 and Fig. 4.

Other points:

1) Please speculate on the underlying mechanism coordinating timing of PB1 ubiquitination at K578.

Response: Our results of the K578A and R mutants strongly suggest that ubiquitination of PB1 is required at an early time point during infection to neutralize the positive charge at PB1-K578 in order to reduce the affinity of the newly produced trimeric and RNA-free polymerase to NP as well as to prevent the assembly of symmetric polymerase dimers, which is required at a later step during vRNA replication. We therefore speculate that PB1-K578 ubiquitination occurs either within the trimeric polymerase or non-assembled PB1 protein, but not the vRNP. However, so far we have no insights whether the modification occurs in the cytoplasm or nucleus and whether it is removed at some later step. Unfortunately, we were not yet able to identify the responsible ubiquitin E3 ligase for K578, which would have enabled us to investigate the question of timing in more detail. Finding answers to these highly important questions will need further intensive experimental analysis.

2) Mutagenesis of identified ubiquitination sites to A and R is well justified. However, even a very conservative K to R exchange can affect protein interactions and/or bioactivity independent of ubiquitination status. Consider that the alleged pro- and antiviral functions

associated with individual residues could alternatively simply reflect a specific requirement for a lysine residue at that particular position.

Response: We thank the reviewer for this thoughtful comment, which we fully support. We have tried to address this common problem by employing a detailed analysis of the individual lysine's structural environment to guide further ideas and investigations (e.g. catalytic residues in PB1 and PA- lines 198-201). However, we agree that based on our analysis we cannot exclude effects related to the biochemical attributes of lysine at a specific position.

3) I am wondering whether the extremely long text blocks in the second half of the results section can be broken down into more palatable sections to make the study more accessible to a broader readership.

Response: In the revised manuscript we have rearranged the previous structure of the manuscript regarding the text as well as the figures in order to put more emphasis on our experimental data, tone down the importance of the in silico analysis and improve the amount of data per section to a better digestible size.

Reviewer #2

General comments

In this manuscript, Guenl et al. aim to identify and characterise ubiquitination sites on the influenza A virus RdRp. The authors begin by using an immunoprecipitation-mass spectrometry approach to identify UB and UBL sites, then follow up with a comprehensive mutational analysis of these sites. The authors use these data to shortlist UB/UBL sites of interest and attempt recombinant virus rescues, eventually focussing on the PB1 K578 amino acid residue. The authors then perform a series of assays to interrogate how K578A and K578R mutations impact RdRp activity and protein-protein interactions, concluding that the K578 residue is important for regulating RdRp dimerization through its ubiquitylation.

The manuscript is clearly written, and most experiments are of high quality. The identification of UB/UBL sites and subsequent mutational analysis is a particular strength of this manuscript. While some of the experiments which investigate the PB1 K578 mutations do produce intriguing results, others appear to suffer from technical issues, and some are mis-interpreted. In addition, the data shown do not support the model presented at the end of the manuscript.

Response: We are thankful to the reviewer for recognizing the importance and quality of the ubiquitination screen and the subsequent functional analysis of the identified lysines in the viral polymerase. We regret that our initial version of the manuscript was not able to convince the reviewer of our final model on how UB modification of PB1-K578 coordinates the position of the N-terminal PB2 loop to adjust NP binding and the assembly of the symmetric dimers in a charge dependent manner.

As the reviewer pointed out that parts of our data may be suffering from technical issues, we repeated several experiments, including all Co-IPs (Figure 6) and the *in silico* analyses (Figure 4) to improve the overall data quality and included also more controls. Based on the reviewer comments we also added some new experiments, such as the cRNA stabilization assay using an inactive PB1 protein (Figure 5c), which provided novel insights and allowed us to refine our final model for the biological function of PB1-K578 ubiquitylation.

We hope that the new structure and data presented in the revised manuscript are more convincing and also justify the final model in the eye of the reviewer.

Please find below a point-by-point response to the reviewer's concerns.

Specific points

1) Fig. 3C, D: For amino acid residues PB2 K482 and PA K22, the alanine/arginine mutations have opposite effects on RdRp activity. How would this make sense if they are UB/UBL acceptor sites as suggested in lines 157-159?

Response:

We agree with the reviewer that based on the classical function of UB, which is to mark proteins for proteasomal degradation, our interpretation of the results from the functional assays would not make a lot of sense for residues PA-K22 and PB2-K482.

However, beyond acting as a signal for proteasomal degradation, UB-modification has a variety of additional effects on the modified lysine itself, which need to be taken into account. One is the neutralization of the positive charge of the side chain by the dipeptide bond, which affects the close structural environment of the modified residue and another is the deposition of a small protein, which itself has different effects: It introduces a binding surface for other

UBL-binding proteins, but can also shield binding sites on the modified protein. We now also included an additional paragraph in the introduction (highlighted in red: lines 85-91)

As we did not observe pronounced differences in the expression levels of the mutated proteins (Supplemental Fig. 2), we excluded that proteasomal degradation is the responsible mechanism for the observed effects.

Instead, we based our interpretation also on previously reported functions of specific residues, e.g. PA-K22 is part of the endonuclease domain and PB2-K482 has been reported as an alternative nuclear localization signal, as well as their near structural environment and putative interactions to other residues.

Therefore, our interpretation of the results for PA-K22 and PB2-K482 is as follows:

PA-K22: is part of the endonuclease domain that is important for viral mRNA transcription and cap snatching. According to Keown, Zhu et al. (2022; PMID: 35017564) it is involved in dynamic interactions with the neighboring residues of PB1. Altering the charge of its sidechain by UBL-modification likely affects its spectrum of interactions.

With regard to the charge of its side chain, **K22A** reflects the lysine in a modified state with a neutral side chain. This neutral state is beneficial for mRNA transcription, evidenced by the increased activity in the polymerase reconstitution assay. We can only speculate about the structural effects or altered interaction to other amino acids that may be involved in this mechanism. We did not elucidate this in more detail. Clearly, we cannot fully exclude that the lack of the UBL-moiety is responsible for the observed effect. However, in that case we would expect that the R mutation would lead to the same result.

K22R inherits a constant positive charge that cannot be neutralized by UB/UBL. For mRNA transcription this scenario seems to be detrimental as polymerase activity is ablated, possibly by fixing the polymerase in a distinct position not allowing for the structural mobility that is required for Cap-Snatching during mRNA transcription. Intriguingly, K22R abolished mRNA transcription but retains the ability of vRNA replication (Fig. 3G), suggesting that the non-modified state of the lysine is required for vRNA replication while the modified state is required to allow mRNA transcription.

For better understanding we included a new text passage (highlighted in red: lines 179-184) as well as an illustration of the location of PA-K22 (Supplemental Fig. 2j) based on the polymerase structures from Keown, Zhu et al.

PB2-K482: this residue is part of an alternative nuclear localization signal (NLS), which are often sensitive to charge alterations but also interact with nuclear transporters to facilitate protein translocation. UBL-modification can therefore have diverse effects. Indeed, we observed that mutations K482R affected the intracellular localization of the PB2 subunit and shifted it from the nucleus into the cytoplasm, while K482A did not affect its localization (Fig. 3L). The underlying mechanism for this cannot be concluded from our analysis. It may be that lysine is required for structural integrity of the NLS or that lack of the UB/UBL alters the interaction to other proteins. From our data we can only conclude that K482R supports mRNA synthesis, while K482A ablates mRNA synthesis, which is in line with previous data from Karim et al. (2020; PMID: 32265326).

However, we would like to point out that further experimental analysis will be needed to reveal the biological mechanism of UB/UBL modification at these specific lysines, which are not in the focus of this manuscript, but may be subject to following studies.

2) Fig. 3I-K: Please include an image of the DAPI channel only, as this stain is not clear in the merged image (such as in Fig. 3K, leftmost panel).

Response: We have now included the DAPI images to the figure (Please see Fig. 3 L-N).

3) Fig. 4H-J: This is another key assay for the conclusions of the manuscript, as the authors use these data to suggest that the PB1 K578 mutations affect both cRNA synthesis and vRNA synthesis. However, these results appear to be misinterpreted. cRNA stabilisation assays normally use an active site mutation in the PB1 subunit to prevent vRNA synthesis, and this allows cRNA synthesis to be examined in isolation (Vreede et al. 2004). In the assay performed here there does not appear to be an active site mutation, so cRNA accumulation will be dependent on vRNA synthesis and vice versa. Therefore, this assay cannot distinguish cRNA synthesis from vRNA synthesis, and rather illustrates the overall efficiency of viral genome replication.

Response: (Please note that these results are now presented in figure 5) We thank the reviewer for this very important comment. We realized that we have indeed referred to this assay incorrectly. As it is performed in the presence of an active PB1 protein, the results of our assay resemble the combined potency of the viral polymerase to stabilize and synthesize cRNA, including the step of vRNA synthesis as correctly pointed out by the reviewer.

To differentiate between cRNA stabilization and the following two steps of RNA synthesis, we have repeated the complete analysis and now also included the inactive PB1 mutant that harbors the mutation D445A/D446A (Vreede et al.). We are now referring to the assay with inactive PB1 as cRNA stabilization (Fig. 5C), while the assays with active PB1 are referred to as cRNA synthesis (Fig. 5D) and vRNA synthesis (Fig. 5E), respectively, to avoid confusion (highlighted in red: lines 337-350). Indeed, this assay now clearly demonstrates that both mutations, PB1-K578A and K578R present WT-like cRNA stabilization activity (Fig. 5C). In contrast, using the active PB1 protein, cRNA synthesis is similarly reduced for both mutants (Fig. 5D). However, vRNA synthesis was only compromised for K578A but not for K578R, suggesting that K578A has suboptimal cRNA and vRNA synthesis activity, while K578R is only compromised in cRNA synthesis in this setting.

4) Fig. 4K: PB1 and PB2 bands are visible in the pulldown sample, but not in the input. Please show the PB1 and PB2 bands in both.

Response: (Please note that these results are now presented in figure 6).

We have repeated all co-IP assays and also included more controls to the assays in Suppl. Fig. 4 A and B. Here, we show that NP interacts with PB2 and to a minor extent also with PB1 in a 1:1 transfection assay. We did not observe NP binding to PA (Suppl. Fig. 4A). We further performed a new experiment to assess whether the mutations PB1-K578A/R affected the interaction between NP and PB1 (Suppl. Fig. 4B) but did not observe any significant difference.

Figure 6A demonstrates NP interaction with the RNA free polymerase trimer. PB1 and PB2 bands are now shown also in the input. We also included the quantification of the normalized western blot signals from five independent replicates as now also indicated in the figure.

Please note that we have also included a **new assay in Figure 6B**. Here, we co-expressed vRNA with the polymerase to assess NP interaction with the vRNA-bound polymerase and found that NP binding in this context is not altered by K578A or R mutations. This suggests that ubiquitination of PB1-K578 reduces the affinity of the free polymerase to NP but not in the context of the vRNP/cRNP.

5) Fig. 4L: This is a key assay for the conclusions of the manuscript; however, the controls do not make sense: Why is Strep-PA not pulled down in the -PB2 control? Why is a significant amount of HA-PA being detected in the pulldown even with no Strep-PA present in the -PB2 control? Why is there a clear Strep-PA band in the input even in the -Strep-PA control?

Response: (Please note that these results are now presented in figure 6).

We apologize for the missing controls and confusing signals. We have repeated all co-IP assays to provide all required controls and correct input protein signals. Please find the results of the repeated assay in **Figure 6C**:

Of note: there is a faint background band for strep-PA in the -strep-PA lane in the input. In addition, we notice some background binding of HA-PA to the beads.

We also included now the quantification of the normalized western blot signals from five independent replicates as provided in the figure.

6) Fig. 4K, L: As these data illustrate a quantitative difference between wild type and K578A, please either show a graph of the quantification with error bars, or include a statement in the legend indicating how many times the experiments have been repeated with similar results.

Response: (Please note that these results are now presented in figure 6).

We have now included graphs with error bars for quantification of the co-IP assays and stated the number of replicates.

7) Fig. 5A: This is a key experiment, as it shows that the modification to PB1 K578 is UB rather than NEDD8. As with Fig. 4K and 4L, this conclusion is based on a quantitative difference, so please either quantify the ubiquitin signal across multiple replicates or include a statement in the legend indicating how many times the experiment has been repeated with similar results.

Response: (Please note that these results are now presented in figure 4A).

We have repeated this assay and now included a quantification with error bars for the ubiquitin signal of the WT protein and the 578A mutant, which was derived from three independent replicates, which we now also state in the figure legend (Fig. 4A). Additionally, we rephrased the part (highlighted in red: lines 230-234)

8) Fig 6E: This model suggests that ubiquitylation of PB1 K578 controls viral genome replication by inhibiting formation of the symmetric dimer. However, this model is not consistent with the dimerization assay shown in Fig. 4L, which demonstrates that the alanine mutation inhibits dimerization while the arginine mutation has no effect.

Surely if ubiquitination of K578 inhibited symmetric dimer formation, both of these mutations would show an increase in dimerization as they cannot be ubiquitylated?

Response: We thank the reviewer for this critical comment and pointing out to us that we have not properly justified our conclusion.

We agree with the reviewer that the effect of ubiquitylation in our model cannot simply be explained by the presence of the UB moiety itself as our data have demonstrated that replacing PB1-K578 with A or R, which both ablate the modification, affect the functionality of the viral polymerase in very different ways. We therefore concluded, that the effect of the modification is not conferred by the UB moiety but by a different mechanism.

Our model takes into account that ubiquitination of a lysine neutralizes the positive charge of the side chain, which is mimicked by the A (charge of lysin when ubiquitylated) and R (charge of lysine without UB) mutations. Combined with the *in silico* analysis, now presented in Fig. 4,

which predicted that A and R mutations at PB1-K578 modulate its structural environment and influence the position of the N-terminal PB2 loop, our data strongly point towards a charge mediated mechanism. The results for polymerase dimerization demonstrated that 578A reduces assembly of the symmetric dimer, while 578R retains WT dimerization levels. Nevertheless, A is tolerated during infection while R is not, suggesting that the affinity of the WT polymerase to assemble the symmetric dimer requires dynamic modulation during infection. Our data indicate that this is mediated in a charge-dependent manner by the acquisition of ubiquitin at lysine 578 in PB1, which controls the freedom of the N-terminal PB2 loop that is part of the interface of the symmetric dimer.

We realized that our model did not appropriately summarize this conclusion. We have therefore included a refined version of our model in the revised manuscript (Fig. 6D). In this new model, we now included a graphical summary of the PB1-K578A and R mutations during viral replication, and also pointed that out in the discussion (highlighted in red: lines 451-466). We hope, that this helps to deliver the main message of our work to the audience more comprehensively.

The fact that the PB2 71-73A mutation has a potent effect in the dimerization assay confirms that this assay is examining the symmetric dimer rather than the asymmetric dimer.

Response: We agree with the reviewer.

9) Supplementals:

Supp. Fig. 2A-F: Please add labels to each panel of the western blots.

Response: We apologize for the inconvenience and have added labels to each panel of the western blots.

10) Supp. Fig. 2A-F: This figure has panels spliced together with more than one wild type lane. Please make it clear which wild type sample should be compared with each of the mutant samples.

Response: We have now split the panels to ensure that the respective wild type samples are shown.

11) Supp. Fig. 2 legend: Panels G-I are mislabelled in the legend.

Response: We apologize for the confusion. We changed the legend labels accordingly.

12) Supp. Fig. 3B: It is not clear whether this is re-blotting samples from the pulldown experiment shown in Fig. 4K, or if it is a separate experiment. Either way, please include all relevant controls for a pulldown assay, such as those included in 4K.

Response: (please note that these results are now presented in supplementary figure 4A and B). We have repeated this experiment and included additional negative controls. Since we also observed a slight interaction between PB1 and NP, we have also included a co-IP to assess whether the PB1-578A/R mutations affect this interaction. However, we could not observe a significant difference in NP binding (Suppl. Fig. 4B)

13) Supp. Fig. 4H legend: Typo, PB2-E72 is repeated twice.

Response: We have corrected this mistake.

14) Table:

Table 1: It would be good to include the virus titres which were obtained from successful rescues.

Response: We included all titres.

Text:

Lines 143-145: From this point on, it is assumed in the text that every amino acid residue identified is modified by ubiquitin. This may not be the case, as the authors can clearly observe NEDDylation of all RdRp subunits in addition to ubiquitylation (Fig. 1A). It should be made clear in the following sections of the manuscript that these could be ubiquitylation or NEDDylation sites.

Response: We thank the reviewer for this hint. The respective text passages were changed to UB/UBL (highlighted in red).

Lines 177-180: This is overly speculative.

Response:

„This suggests that neither a constantly neutral nor a positively charged residue is tolerated at this position and indicates a requirement for a lysine as a UB acceptor site to achieve neutralization of the positive charge at a distinct time point during viral replication.“

Was changed to:

„Residue PA-K536, which resides within a putative RNA template binding groove 39,40 demonstrated immediate reversion to K536 during virus rescue, demonstrating strong selection pressure for lysine. We speculate, that this indicates that neither a constantly neutral nor a positively charged residue is tolerated at this position and that at some point during viral replication, the positive charge of K536 needs to be neutralized, hypothetically by the addition of UB/UBL.“ (highlighted in red: lines 192 – 197)

Lines 313-318: The statement that the PB1 K578R mutation promotes the symmetric dimer interaction is not consistent with the dimerization assay in Fig. 4L, which shows that K578R has no effect.

Response: (please note that the Co-IPs are now presented in Figure 6)

We agree with the reviewer that the word “promotes” was not well chosen to describe the effect of the 578R mutation. As we have repeated the *in silico* analysis, this part has now been carefully rewritten (please see lines 262-315 in the revised manuscript).

The results of the Co-IPs in Fig. 4L (now figure 6C) show that the formation of the symmetric dimer depends on a positive charge at position 578 in PB1. The positive charge can be provided by the natural lysine (K578) or arginine (K578R) mutation, therefore both proteins demonstrate the same level of symmetric dimer formation in the experiment.

While the results of the Co-IPs show no differences between WT K578 and mutant K578R, the new results of the *in silico* analysis predicted that K578R, in contrast to K578, fixes the PB2 loop in a distinct position, which we assume represents the position of the loop required for assembly of the symmetric dimer. While K/R result in similar levels of dimerization in the

transfection assay, the consequences of a constant positive charge during infection are deleterious (as demonstrated by the high pressure of the R mutant to revert to K).

Please remove speculative conclusions and tone down the abstract.

Response: We have revised the manuscript including and toned down the abstract.

Reviewer #3 (Remarks to the Author):

The authors studied the ubiquitination of polymerase of influenza A virus by the host E3 ligase during infection. They identified that mutations of PB1-K578A and K578R which prevented ubiquitination by E3 increased the polymerase activity, albeit having an adverse effect on cRNP stabilization. The authors used molecular dynamics simulations to explain the effect of the mutations concluding that the charge neutralization of K578 caused by ubiquitination facilitates viral replication by increasing the flexibility of PB2 loop.

The results from the molecular dynamics simulations lack the clarity and statistical analysis to support the conclusions drawn. More specifically, the “mobility” of PB2 loop is not directly analyzed. Several important details of the methodology are missing, compromising the reproducibility of the results.

Response: The reviewer makes a good point and we followed this advice. We revised the *in silico* data section and extended the methodology description. In agreement with Reviewer #1 the analyses were toned down and simplified. Simulations were repeated to a total number of $n=4$ for each condition (WT, K578A, K578R). All results were statistically analyzed by Welch corrected t-test. We also restructured the manuscript and repositioned the *in silico* analysis to the beginning of the functional analysis with the aim to emphasize its predictive character. Please find the description of the results of the new *in silico* analysis in lines 262-315 (highlighted in red).

Below are concerns specific to the section “PB1-K578 controls the spatio-temporal position of the PB2 loop in a charge-dependent manner:”

1. “This analysis further indicated that K578R resulted in prolonged and less dynamic interaction with residues PB2-Q73 and R101 compared to K578 (Fig. 6C and D), which likely leads to a stabilization of the PB2 loop and thereby promotes the interaction to the trans-activating polymerase that is required for the formation of the symmetric dimer”

- to support this statement, a dynamic estimate of the interaction should be presented. (e.g. lifetime of the interactions)

Response: We revised the complete section and analyzed the conformational differences caused by mutation using RMSD distribution over the complete simulation time. Please find the results of the new analysis in Figure 4.

2. “These results suggest that the defect in vRNA replication of the K578R mutant is caused by restricted mobility of the PB2 loop and thereby constitutes the selection pressure for reversion to K578. Presumably, neutralization of K578 by ubiquitination provides a molecular switch that determines the stability of the PB2 loop, which facilitates transition of the transcriptase to the replicase and regulates both steps of vRNA replication (Fig 6E).”

- the mobility of the PB2 loop was not analyzed and the changes in the secondary structure shown in fig 6A do not immediately lead to this conclusion. We suggest demonstrating the difference in fluctuations of the PB2 as a more appropriate quantity.

Response: We now present the mean root mean square fluctuation (RMSF) for each condition (WT, K578A, K578R) and analyzed the results statistically ($n=4$ for each condition). K578A significantly alters the RMSF of distinct PB2 loop residues.

Below are concerns related to the methodology and presentation of the results:

1. The confidence interval and statistical significance of the measured quantities (secondary structure in fig 6A and quantities S4I) is not reported. A comparison of the mean values (with confidence intervals and statistical significance) is desired as well as a larger size/resolution of the panel with a focus/highlight of the region of interest.

Response: (please note that the results of the new *in silico* analysis are presented in figure 4 of the revised manuscript)

We repeated 100 ns MD simulations to a total number of four independent simulations for each condition (WT, K578R, K578A). Means \pm SEM and significance of mean differences analyzed by Welch corrected t-test are given for analyzed quantities.

2. In Fig 6B-D, It is not clear what is the purpose of presenting the intersection of measured “ionic” “hydrophobic” and other quantities, rather than only the quantities themselves. We suggest either removing them or discussing their significance in the text.

Response: We revised the section and focused our interpretation only on the *in vitro* predicted ionic interaction between PB2 residue E72 and its interaction partners.

3. “The complex interactions were analyzed in detail using Yasara ligand” - which interactions specifically and in what details? Hydrogen bonds can be formed between the sidechains as well as the backbone of the residue (which would be independent of the mutation). We suggest the authors rewrite this statement or at least specify which moiety is involved in the interaction.

Response: We revised the section accordingly. In agreement with Reviewer #1 we toned down the interpretation and focused only on the *in vitro* predicted ionic interaction between PB2 residue E72 and its interaction partners. Ionic interactions were formed between the deprotonated carboxy-group of E72 and the corresponding protonated site chains of K578, R101, R571 and R572.

4. The rationale for conducting “local” simulations is not explained.

Response: We added a short explanation to the results and materials as well as the methods section. Based on the *in vitro* data as well as the recent literature we were able to specify the region of interest. Due to the extensive structure of the trimeric polymerase, all atoms mobile simulations of the complete protein complex would need a tremendous amount of computational resources even for a much shorter simulation time than 100 ns. Therefore, we restricted the all atoms mobile simulation in agreement with the previous *in vitro* data to the extended PB1-PB2 loop interface. This is now explained in the text: please see lines 267 – 268.

5. Q73 is not labeled in fig. 6A. The coloring of the labeled residues does not appear to correspond to the legend. How were these colors chosen and why?

Response: Due to the new structure of the manuscript Figure 6A is now Figure 4B. We have now labeled Q73 as well as PB1-R571 and R572.

6. The following details are missing from the methods section:

The method section for Molecular dynamic simulations was completely revised (highlighted in red: lines 976-1002).

a. What was the forcefield used for nucleic acid?

Response: Homology model was built from structure including a bound nucleic acid molecule. However, atoms of nucleic acids were immobilized for all models before local MD simulations were performed. Local 100 ns simulations were conducted containing only protein residues

using AMBER15IPQ force field. For clarification, we added the following sentence to the materials and method section: “Atoms in a 45Å-sphere around residue PB1-K578 including only protein residues were set mobile, while all other atoms were immobilized”.

b. Reference to the AMBER forcefield missing

Response: We added the reference to the AMBER15IPQ force field.

c. What temperature was the simulation conducted at and what was the algorithm used to control it? Same for pressure

Response: Simulation was performed at 298 K and 1 atm. Simulation temperature and pressure were constantly controlled by NPT ensemble previously described in detail by Krieger et al 2015 (Krieger, E. & Vriend, G. New ways to boost molecular dynamics simulations. Journal of Computational Chemistry 36, 996–1007 (2015)) The reference was added to the materials and method section.

d. What was the solvent model used?

Response: TIP3P water model was used for the solvent and is mentioned in the materials and methods section.

e. What was the integrator in the simulations?

Response: Atomic motions were integrated with a multiple timestep of 2x1.25 fs for bonded interactions and 2.5 fs for non-bonded interactions as previously described by Krieger et al 2015 (Krieger, E. & Vriend, G. New ways to boost molecular dynamics simulations. Journal of Computational Chemistry 36, 996–1007 (2015)).

f. How was the protonation of residues determined?

Response: Protonation state was determined by initialization procedure implemented in YASARA structure (vers. 21.8.27). A reference including detailed description of the process is added to materials and methods section.

g. How were the adjacent residues to K/A/R578 (R101, E72 and Q73) determined?

Response: We revised the *in silico* analyses section and focused on interactions with residues that were identified as potential interaction partners by data of *in vitro* experiments.

h. How was the secondary structure “abundance” calculated?

Response: In agreement with the comments of reviewer #1 we reduced the *in silico* analyses and focused it on the influence of ionic interactions between E72 and the surrounding residues. Therefore, secondary structure analyses were removed.

REVIEWER COMMENTS

Reviewer #2 (Remarks to the Author):

Please find the review comments in red letter in the attached document.

Point-by-point response to the reviewer remarks

Reviewer #1

This study applied a mass spec approach to map the ubiquitination landscape of the IAV polymerase complex. In the first half of the manuscript, authors provide an overview of mass spec results, nicely presenting the data in a structural context, and mechanistically characterize identified residues through a mutagenesis follow-up. Novel sites with, when ubiquitinated, proviral and antiviral activity were convincingly identified. The second half of the manuscript focusses on residue K578 in the PB1 thumb domain specifically, proposing ubiquitination at this position as a regulator of polymerase dimerization and vRNA replication. These are important findings that represent a major advance of the understanding of the regulation of influenza virus polymerase activity. Data are overall of high quality and support the major conclusions. However, *in silico* always delivers, especially when based on homology models. The presentation of homology model predictions and molecular dynamics simulations needs to be toned down, even if the models created are very persuasive.

Response: We thank the reviewer for this overall very positive comment on our ubiquitination screen and functional analysis of the modified lysines in the viral polymerase as well as the appreciation of the novelty and importance of our results on K578 modification for the general understanding of this complex enzyme. We understand that we have in parts overstated the results of the *in silico* analysis. We have now completely redone the *in silico* analysis to provide statistical solidity of our data and refined insights into the interplay of residues PB1-K578 and PB2-E72. This has resulted in novel insight into how the position of the PB2 loop is coordinated (please see new Fig. 4.) In addition, instead of reporting the *in silico* results at the end of the results part, we repositioned the newly written part on the *in silico* analysis before the further experimental analysis and now clearly state in the text that this analysis is a prediction. We hope that this meets the reviewer's concerns. Please find the newly written part and the corresponding figures on the *in silico* analysis in lines 262-315 and Fig. 4.

Other points:

1) Please speculate on the underlying mechanism coordinating timing of PB1 ubiquitination at K578.

Response: Our results of the K578A and R mutants strongly suggest that ubiquitination of PB1 is required at an early time point during infection to neutralize the positive charge at PB1-K578 in order to reduce the affinity of the newly produced trimeric and RNA-free polymerase to NP as well as to prevent the assembly of symmetric polymerase dimers, which is required at a later step during vRNA replication. We therefore speculate that PB1-K578 ubiquitination occurs either within the trimeric polymerase or non-assembled PB1 protein, but not the vRNP. However, so far we have no insights whether the modification occurs in the cytoplasm or nucleus and whether it is removed at some later step. Unfortunately, we were not yet able to identify the responsible ubiquitin E3 ligase for K578, which would have enabled us to investigate the question of timing in more detail. Finding answers to these highly important questions will need further intensive experimental analysis.

2) Mutagenesis of identified ubiquitination sites to A and R is well justified. However, even a very conservative K to R exchange can affect protein interactions and/or bioactivity independent of ubiquitination status. Consider that the alleged pro- and antiviral functions

associated with individual residues could alternatively simply reflect a specific requirement for a lysine residue at that particular position.

Response: We thank the reviewer for this thoughtful comment, which we fully support. We have tried to address this common problem by employing a detailed analysis of the individual lysine's structural environment to guide further ideas and investigations (e.g. catalytic residues in PB1 and PA- lines 198-201). However, we agree that based on our analysis we cannot exclude effects related to the biochemical attributes of lysine at a specific position.

3) I am wondering whether the extremely long text blocks in the second half of the results section can be broken down into more palatable sections to make the study more accessible to a broader readership.

Response: In the revised manuscript we have rearranged the previous structure of the manuscript regarding the text as well as the figures in order to put more emphasis on our experimental data, tone down the importance of the in silico analysis and improve the amount of data per section to a better digestible size.

Reviewer #2

General comments

In this manuscript, Guenl et al. aim to identify and characterise ubiquitination sites on the influenza A virus RdRp. The authors begin by using an immunoprecipitation-mass spectrometry approach to identify UB and UBL sites, then follow up with a comprehensive mutational analysis of these sites. The authors use these data to shortlist UB/UBL sites of interest and attempt recombinant virus rescues, eventually focussing on the PB1 K578 amino acid residue. The authors then perform a series of assays to interrogate how K578A and K578R mutations impact RdRp activity and protein-protein interactions, concluding that the K578 residue is important for regulating RdRp dimerization through its ubiquitylation.

The manuscript is clearly written, and most experiments are of high quality. The identification of UB/UBL sites and subsequent mutational analysis is a particular strength of this manuscript. While some of the experiments which investigate the PB1 K578 mutations do produce intriguing results, others appear to suffer from technical issues, and some are mis-interpreted. In addition, the data shown do not support the model presented at the end of the manuscript.

Response: We are thankful to the reviewer for recognizing the importance and quality of the ubiquitination screen and the subsequent functional analysis of the identified lysines in the viral polymerase. We regret that our initial version of the manuscript was not able to convince the reviewer of our final model on how UB modification of PB1-K578 coordinates the position of the N-terminal PB2 loop to adjust NP binding and the assembly of the symmetric dimers in a charge dependent manner.

As the reviewer pointed out that parts of our data may be suffering from technical issues, we repeated several experiments, including all Co-IPs (Figure 6) and the *in silico* analyses (Figure 4) to improve the overall data quality and included also more controls. Based on the reviewer comments we also added some new experiments, such as the cRNA stabilization assay using an inactive PB1 protein (Figure 5c), which provided novel insights and allowed us to refine our final model for the biological function of PB1-K578 ubiquitylation.

We hope that the new structure and data presented in the revised manuscript are more convincing and also justify the final model in the eye of the reviewer.

Please find below a point-by-point response to the reviewer's concerns.

No further comment.

Specific points

1) Fig. 3C, D: For amino acid residues PB2 K482 and PA K22, the alanine/arginine mutations have opposite effects on RdRp activity. How would this make sense if they are UB/UBL acceptor sites as suggested in lines 157-159?

Response:

We agree with the reviewer that based on the classical function of UB, which is to mark proteins for proteasomal degradation, our interpretation of the results from the functional assays would not make a lot of sense for residues PA-K22 and PB2-K482.

However, beyond acting as a signal for proteasomal degradation, UB-modification has a variety of additional effects on the modified lysine itself, which need to be taken into account. One is the neutralization of the positive charge of the side chain by the dipeptide bond, which

affects the close structural environment of the modified residue and another is the deposition of a small protein, which itself has different effects: It introduces a binding surface for other UBL-binding proteins, but can also shield binding sites on the modified protein. We now also included an additional paragraph in the introduction (highlighted in red: lines 85-91)

As we did not observe pronounced differences in the expression levels of the mutated proteins (Supplemental Fig. 2), we excluded that proteasomal degradation is the responsible mechanism for the observed effects.

Instead, we based our interpretation also on previously reported functions of specific residues, e.g. PA-K22 is part of the endonuclease domain and PB2-K482 has been reported as an alternative nuclear localization signal, as well as their near structural environment and putative interactions to other residues.

Therefore, our interpretation of the results for PA-K22 and PB2-K482 is as follows:

PA-K22: is part of the endonuclease domain that is important for viral mRNA transcription and cap snatching. According to Keown, Zhu et al. (2022; PMID: 35017564) it is involved in dynamic interactions with the neighboring residues of PB1. Altering the charge of its sidechain by UBL-modification likely affects its spectrum of interactions.

With regard to the charge of its side chain, **K22A** reflects the lysine in a modified state with a neutral side chain. This neutral state is beneficial for mRNA transcription, evidenced by the increased activity in the polymerase reconstitution assay. We can only speculate about the structural effects or altered interaction to other amino acids that may be involved in this mechanism. We did not elucidate this in more detail. Clearly, we cannot fully exclude that the lack of the UBL-moiety is responsible for the observed effect. However, in that case we would expect that the R mutation would lead to the same result.

K22R inherits a constant positive charge that cannot be neutralized by UB/UBL. For mRNA transcription this scenario seems to be detrimental as polymerase activity is ablated, possibly by fixing the polymerase in a distinct position not allowing for the structural mobility that is required for Cap-Snatching during mRNA transcription. Intriguingly, K22R abolished mRNA transcription but retains the ability of vRNA replication (Fig. 3G), suggesting that the non-modified state of the lysine is required for vRNA replication while the modified state is required to allow mRNA transcription.

For better understanding we included a new text passage (highlighted in red: lines 179-184) as well as an illustration of the location of PA-K22 (Supplemental Fig. 2j) based on the polymerase structures from Keown, Zhu et al.

PB2-K482: this residue is part of an alternative nuclear localization signal (NLS), which are often sensitive to charge alterations but also interact with nuclear transporters to facilitate protein translocation. UBL-modification can therefore have diverse effects. Indeed, we observed that mutations K482R affected the intracellular localization of the PB2 subunit and shifted it from the nucleus into the cytoplasm, while K482A did not affect its localization (Fig. 3L). The underlying mechanism for this cannot be concluded from our analysis. It may be that lysine is required for structural integrity of the NLS or that lack of the UB/UBL alters the interaction to other proteins. From our data we can only conclude that K482R supports mRNA synthesis, while K482A ablates mRNA synthesis, which is in line with previous data from Karim et al. (2020; PMID: 32265326).

However, we would like to point out that further experimental analysis will be needed to reveal the biological mechanism of UB/UBL modification at these specific lysines, which are not in the focus of this manuscript, but may be subject to following studies.

No further comment.

2) Fig. 3I-K: Please include an image of the DAPI channel only, as this stain is not clear in the merged image (such as in Fig. 3K, leftmost panel).

Response: We have now included the DAPI images to the figure (Please see Fig. 3 L-N).

No further comment.

3) Fig. 4H-J: This is another key assay for the conclusions of the manuscript, as the authors use these data to suggest that the PB1 K578 mutations affect both cRNA synthesis and vRNA synthesis. However, these results appear to be misinterpreted. cRNA stabilisation assays normally use an active site mutation in the PB1 subunit to prevent vRNA synthesis, and this allows cRNA synthesis to be examined in isolation (Vreede et al. 2004). In the assay performed here there does not appear to be an active site mutation, so cRNA accumulation will be dependent on vRNA synthesis and vice versa. Therefore, this assay cannot distinguish cRNA synthesis from vRNA synthesis, and rather illustrates the overall efficiency of viral genome replication.

Response: (Please note that these results are now presented in figure 5) We thank the reviewer for this very important comment. We realized that we have indeed referred to this assay incorrectly. As it is performed in the presence of an active PB1 protein, the results of our assay resemble the combined potency of the viral polymerase to stabilize and synthesize cRNA, including the step of vRNA synthesis as correctly pointed out by the reviewer.

To differentiate between cRNA stabilization and the following two steps of RNA synthesis, we have repeated the complete analysis and now also included the inactive PB1 mutant that harbors the mutation D445A/D446A (Vreede et al.). We are now referring to the assay with inactive PB1 as cRNA stabilization (Fig. 5C), while the assays with active PB1 are referred to as cRNA synthesis (Fig. 5D) and vRNA synthesis (Fig. 5E), respectively, to avoid confusion (highlighted in red: lines 337-350). Indeed, this assay now clearly demonstrates that both mutations, PB1-K578A and K578R present WT-like cRNA stabilization activity (Fig. 5C). In contrast, using the active PB1 protein, cRNA synthesis is similarly reduced for both mutants (Fig. 5D). However, vRNA synthesis was only compromised for K578A but not for K578R, suggesting that K578A has suboptimal cRNA and vRNA synthesis activity, while K578R is only compromised in cRNA synthesis in this setting.

No further comment.

4) Fig. 4K: PB1 and PB2 bands are visible in the pulldown sample, but not in the input. Please show the PB1 and PB2 bands in both.

Response: (Please note that these results are now presented in figure 6).

We have repeated all co-IP assays and also included more controls to the assays in Suppl. Fig. 4 A and B. Here, we show that NP interacts with PB2 and to a minor extent also with PB1 in a 1:1 transfection assay. We did not observe NP binding to PA (Suppl. Fig. 4A). We further performed a new experiment to assessed whether the mutations PB1-K578A/R affected the interaction between NP and PB1 (Suppl. Fig. 4B) but did not observe any significant difference.

Figure 6A demonstrates NP interaction with the RNA free polymerase trimer. PB1 and PB2 bands are now shown also in the input. We also included the quantification of the normalized western blot signals from five independent replicates as now also indicated in the figure. Please note that we have also included a **new assay in Figure 6B**. Here, we co-expressed vRNA with the polymerase to assess NP interaction with the vRNA-bound polymerase and found that NP binding in this context is not altered by K578A or R mutations. This suggests that ubiquitination of PB1-K578 reduces the affinity of the free polymerase to NP but not in the context of the vRNP/cRNP.

The authors have made a substantial effort to improve the quality of all pulldown assays in the revised manuscript, however, the interpretation of some of the data remains problematic. Specifically, in Figure 6a the authors observe reduced HA-PA signal in the pulldown using PB1 K578A, but the levels of PB1 and PB2 in the pulldown are equal to wild type. The authors interpret this result as a reduced NP-RdRp interaction, but if this were the case then PB1 and PB2 would also be reduced in the pulldown, which they are not.

The fact that only PA is reduced in the pulldown instead indicates that the PB1 K578A mutation is preventing PA from interacting properly with PB1 and PB2; in other words, the mutation prevents the trimeric RdRp from assembling correctly.

5) Fig. 4L: This is a key assay for the conclusions of the manuscript; however, the controls do not make sense: Why is Strep-PA not pulled down in the -PB2 control? Why is a significant amount of HA-PA being detected in the pulldown even with no Strep-PA present in the -PB2 control? Why is there a clear Strep-PA band in the input even in the -Strep-PA control?

Response: (Please note that these results are now presented in figure 6).

We apologize for the missing controls and confusing signals. We have repeated all co-IP assays to provide all required controls and correct input protein signals. Please find the results of the repeated assay in **Figure 6C**:

Of note: there is a faint background band for strep-PA in the -strep-PA lane in the input. In addition, we notice some background binding of HA-PA to the beads.

We also included now the quantification of the normalized western blot signals from five independent replicates as provided in the figure.

No further comment.

6) Fig. 4K, L: As these data illustrate a quantitative difference between wild type and K578A, please either show a graph of the quantification with error bars, or include a statement in the legend indicating how many times the experiments have been repeated with similar results.

Response: (Please note that these results are now presented in figure 6).

We have now included graphs with error bars for quantification of the co-IP assays and stated the number of replicates.

No further comment.

7) Fig. 5A: This is a key experiment, as it shows that the modification to PB1 K578 is UB rather than NEDD8. As with Fig. 4K and 4L, this conclusion is based on a quantitative difference, so

please either quantify the ubiquitin signal across multiple replicates or include a statement in the legend indicating how many times the experiment has been repeated with similar results.

Response: (Please note that these results are now presented in figure 4A).

We have repeated this assay and now included a quantification with error bars for the ubiquitin signal of the WT protein and the 578A mutant, which was derived from three independent replicates, which we now also state in the figure legend (Fig. 4A). Additionally, we rephrased the part (highlighted in red: lines 230-234)

No further comment.

8) Fig 6E: This model suggests that ubiquitylation of PB1 K578 controls viral genome replication by inhibiting formation of the symmetric dimer. However, this model is not consistent with the dimerization assay shown in Fig. 4L, which demonstrates that the alanine mutation inhibits dimerization while the arginine mutation has no effect.

Surely if ubiquitination of K578 inhibited symmetric dimer formation, both of these mutations would show an increase in dimerization as they cannot be ubiquitylated?

Response: We thank the reviewer for this critical comment and pointing out to us that we have not properly justified our conclusion.

We agree with the reviewer that the effect of ubiquitylation in our model cannot simply be explained by the presence of the UB moiety itself as our data have demonstrated that replacing PB1-K578 with A or R, which both ablate the modification, affect the functionality of the viral polymerase in very different ways. We therefore concluded, that the effect of the modification is not conferred by the UB moiety but by a different mechanism.

Our model takes into account that ubiquitination of a lysine neutralizes the positive charge of the side chain, which is mimicked by the A (charge of lysine when ubiquitylated) and R (charge of lysine without UB) mutations. Combined with the *in silico* analysis, now presented in Fig. 4, which predicted that A and R mutations at PB1-K578 modulate its structural environment and influence the position of the N-terminal PB2 loop, our data strongly point towards a charge mediated mechanism. The results for polymerase dimerization demonstrated that 578A reduces assembly of the symmetric dimer, while 578R retains WT dimerization levels. Nevertheless, A is tolerated during infection while R is not, suggesting that the affinity of the WT polymerase to assemble the symmetric dimer requires dynamic modulation during infection. Our data indicate that this is mediated in a charge-dependent manner by the acquisition of ubiquitin at lysine 578 in PB1, which controls the freedom of the N-terminal PB2 loop that is part of the interface of the symmetric dimer.

We realized that our model did not appropriately summarize this conclusion. We have therefore included a refined version of our model in the revised manuscript (Fig. 6D). In this new model, we now included a graphical summary of the PB1-K578A and R mutations during viral replication, and also pointed that out in the discussion (highlighted in red: lines 451-466). We hope, that this helps to deliver the main message of our work to the audience more comprehensively.

The fact that the PB2 71-73A mutation has a potent effect in the dimerization assay confirms that this assay is examining the symmetric dimer rather than the asymmetric dimer.

Response: We agree with the reviewer.

No further comment.

9) Supplementals:

Supp. Fig. 2A-F: Please add labels to each panel of the western blots.

Response: We apologize for the inconvenience and have added labels to each panel of the western blots.

No further comment.

10) Supp. Fig. 2A-F: This figure has panels spliced together with more than one wild type lane. Please make it clear which wild type sample should be compared with each of the mutant samples.

Response: We have now split the panels to ensure that the respective wild type samples are shown.

No further comment.

11) Supp. Fig. 2 legend: Panels G-I are mislabelled in the legend.

Response: We apologize for the confusion. We changed the legend labels accordingly.

No further comment.

12) Supp. Fig. 3B: It is not clear whether this is re-blotting samples from the pulldown experiment shown in Fig. 4K, or if it is a separate experiment. Either way, please include all relevant controls for a pulldown assay, such as those included in 4K.

Response: (please note that these results are now presented in supplementary figure 4A and B). We have repeated this experiment and included additional negative controls. Since we also observed a slight interaction between PB1 and NP, we have also included a co-IP to assess whether the PB1-578A/R mutations affect this interaction. However, we could not observe a significant difference in NP binding (Suppl. Fig. 4B)

No further comment.

13) Supp. Fig. 4H legend: Typo, PB2-E72 is repeated twice.

Response: We have corrected this mistake.

No further comment.

14) Table:

Table 1: It would be good to include the virus titres which were obtained from successful rescues.

Response: We included all titres.

No further comment.

Text:

Lines 143-145: From this point on, it is assumed in the text that every amino acid residue identified is modified by ubiquitin. This may not be the case, as the authors can clearly observe NEDDylation of all RdRp subunits in addition to ubiquitylation (Fig. 1A). It should be made

clear in the following sections of the manuscript that these could be ubiquitylation or NEDDylation sites.

Response: We thank the reviewer for this hint. The respective text passages were changed to UB/UBL (highlighted in red).

No further comment.

Lines 177-180: This is overly speculative.

Response:

„This suggests that neither a constantly neutral nor a positively charged residue is tolerated at this position and indicates a requirement for a lysine as a UB acceptor site to achieve neutralization of the positive charge at a distinct time point during viral replication.“

Was changed to:

„Residue PA-K536, which resides within a putative RNA template binding groove 39,40 demonstrated immediate reversion to K536 during virus rescue, demonstrating strong selection pressure for lysine. We speculate, that this indicates that neither a constantly neutral nor a positively charged residue is tolerated at this position and that at some point during viral replication, the positive charge of K536 needs to be neutralized, hypothetically by the addition of UB/UBL.“ (highlighted in red: lines 192 – 197)

No further comment.

Lines 313-318: The statement that the PB1 K578R mutation promotes the symmetric dimer interaction is not consistent with the dimerization assay in Fig. 4L, which shows that K578R has no effect.

Response: (please note that the Co-IPs are now presented in Figure 6)

We agree with the reviewer that the word “promotes” was not well chosen to describe the effect of the 578R mutation. As we have repeated the *in silico* analysis, this part has now been carefully rewritten (please see lines 262-315 in the revised manuscript).

The results of the Co-IPs in Fig. 4L (now figure 6C) show that the formation of the symmetric dimer depends on a positive charge at position 578 in PB1. The positive charge can be provided by the natural lysine (K578) or arginine (K578R) mutation, therefore both proteins demonstrate the same level of symmetric dimer formation in the experiment.

While the results of the Co-IPs show no differences between WT K578 and mutant K578R, the new results of the *in silico* analysis predicted that K578R, in contrast to K578, fixes the PB2 loop in a distinct position, which we assume represents the position of the loop required for assembly of the symmetric dimer. While K/R result in similar levels of dimerization in the transfection assay, the consequences of a constant positive charge during infection are deleterious (as demonstrated by the high pressure of the R mutant to revert to K).

No further comment.

Please remove speculative conclusions and tone down the abstract.

Response: We have revised the manuscript including and toned down the abstract.

No further comment.

Reviewer #3 (Remarks to the Author):

The authors studied the ubiquitination of polymerase of influenza A virus by the host E3 ligase during infection. They identified that mutations of PB1-K578A and K578R which prevented ubiquitination by E3 increased the polymerase activity, albeit having an adverse effect on cRNP stabilization. The authors used molecular dynamics simulations to explain the effect of the mutations concluding that the charge neutralization of K578 caused by ubiquitination facilitates viral replication by increasing the flexibility of PB2 loop.

The results from the molecular dynamics simulations lack the clarity and statistical analysis to support the conclusions drawn. More specifically, the “mobility” of PB2 loop is not directly analyzed. Several important details of the methodology are missing, compromising the reproducibility of the results.

Response: The reviewer makes a good point and we followed this advice. We revised the *in silico* data section and extended the methodology description. In agreement with Reviewer #1 the analyses were toned down and simplified. Simulations were repeated to a total number of $n=4$ for each condition (WT, K578A, K578R). All results were statistically analyzed by Welch corrected t-test. We also restructured the manuscript and repositioned the *in silico* analysis to the beginning of the functional analysis with the aim to emphasize its predictive character. Please find the description of the results of the new *in silico* analysis in lines 262-315 (highlighted in red).

Below are concerns specific to the section “PB1-K578 controls the spatio-temporal position of the PB2 loop in a charge-dependent manner:”

1. “This analysis further indicated that K578R resulted in prolonged and less dynamic interaction with residues PB2-Q73 and R101 compared to K578 (Fig. 6C and D), which likely leads to a stabilization of the PB2 loop and thereby promotes the interaction to the trans-activating polymerase that is required for the formation of the symmetric dimer”

- to support this statement, a dynamic estimate of the interaction should be presented. (e.g. lifetime of the interactions)

Response: We revised the complete section and analyzed the conformational differences caused by mutation using RMSD distribution over the complete simulation time. Please find the results of the new analysis in Figure 4.

2. “These results suggest that the defect in vRNA replication of the K578R mutant is caused by restricted mobility of the PB2 loop and thereby constitutes the selection pressure for reversion to K578. Presumably, neutralization of K578 by ubiquitination provides a molecular switch that determines the stability of the PB2 loop, which facilitates transition of the transcriptase to the replicase and regulates both steps of vRNA replication (Fig 6E).”

- the mobility of the PB2 loop was not analyzed and the changes in the secondary structure shown in fig 6A do not immediately lead to this conclusion. We suggest demonstrating the difference in fluctuations of the PB2 as a more appropriate quantity.

Response: We now present the mean root mean square fluctuation (RMSF) for each condition (WT, K578A, K578R) and analyzed the results statistically ($n=4$ for each condition). K578A significantly alters the RMSF of distinct PB2 loop residues.

Below are concerns related to the methodology and presentation of the results:

1. The confidence interval and statistical significance of the measured quantities (secondary structure in fig 6A and quantities S4I) is not reported. A comparison of the mean values (with confidence intervals and statistical significance) is desired as well as a larger size/resolution of the panel with a focus/highlight of the region of interest.

Response: (please note that the results of the new *in silico* analysis are presented in figure 4 of the revised manuscript)

We repeated 100 ns MD simulations to a total number of four independent simulations for each condition (WT, K578R, K578A). Means \pm SEM and significance of mean differences analyzed by Welch corrected t-test are given for analyzed quantities.

2. In Fig 6B-D, It is not clear what is the purpose of presenting the intersection of measured “ionic” “hydrophobic” and other quantities, rather than only the quantities themselves. We suggest either removing them or discussing their significance in the text.

Response: We revised the section and focused our interpretation only on the *in vitro* predicted ionic interaction between PB2 residue E72 and its interaction partners.

3. “The complex interactions were analyzed in detail using Yasara ligand” - which interactions specifically and in what details? Hydrogen bonds can be formed between the sidechains as well as the backbone of the residue (which would be independent of the mutation). We suggest the authors rewrite this statement or at least specify which moiety is involved in the interaction.

Response: We revised the section accordingly. In agreement with Reviewer #1 we toned down the interpretation and focused only on the *in vitro* predicted ionic interaction between PB2 residue E72 and its interaction partners. Ionic interactions were formed between the deprotonated carboxy-group of E72 and the corresponding protonated site chains of K578, R101, R571 and R572.

4. The rationale for conducting “local” simulations is not explained.

Response: We added a short explanation to the results and materials as well as the methods section. Based on the *in vitro* data as well as the recent literature we were able to specify the region of interest. Due to the extensive structure of the trimeric polymerase, all atoms mobile simulations of the complete protein complex would need a tremendous amount of computational resources even for a much shorter simulation time than 100 ns. Therefore, we restricted the all atoms mobile simulation in agreement with the previous *in vitro* data to the extended PB1-PB2 loop interface. This is now explained in the text: please see lines 267 – 268.

5. Q73 is not labeled in fig. 6A. The coloring of the labeled residues does not appear to correspond to the legend. How were these colors chosen and why?

Response: Due to the new structure of the manuscript Figure 6A is now Figure 4B. We have now labeled Q73 as well as PB1-R571 and R572.

6. The following details are missing from the methods section:

The method section for Molecular dynamic simulations was completely revised (highlighted in red: lines 976-1002).

a. What was the forcefield used for nucleic acid?

Response: Homology model was built from structure including a bound nucleic acid molecule. However, atoms of nucleic acids were immobilized for all models before local MD simulations were performed. Local 100 ns simulations were conducted containing only protein residues

using AMBER15IPQ force field. For clarification, we added the following sentence to the materials and method section: “Atoms in a 45Å-sphere around residue PB1-K578 including only protein residues were set mobile, while all other atoms were immobilized”.

b. Reference to the AMBER forcefield missing

Response: We added the reference to the AMBER15IPQ force field.

c. What temperature was the simulation conducted at and what was the algorithm used to control it? Same for pressure

Response: Simulation was performed at 298 K and 1 atm. Simulation temperature and pressure were constantly controlled by NPT ensemble previously described in detail by Krieger et al 2015 (Krieger, E. & Vriend, G. New ways to boost molecular dynamics simulations. Journal of Computational Chemistry 36, 996–1007 (2015)) The reference was added to the materials and method section.

d. What was the solvent model used?

Response: TIP3P water model was used for the solvent and is mentioned in the materials and methods section.

e. What was the integrator in the simulations?

Response: Atomic motions were integrated with a multiple timestep of 2x1.25 fs for bonded interactions and 2.5 fs for non-bonded interactions as previously described by Krieger et al 2015 (Krieger, E. & Vriend, G. New ways to boost molecular dynamics simulations. Journal of Computational Chemistry 36, 996–1007 (2015)).

f. How was the protonation of residues determined?

Response: Protonation state was determined by initialization procedure implemented in YASARA structure (vers. 21.8.27). A reference including detailed description of the process is added to materials and methods section.

g. How were the adjacent residues to K/A/R578 (R101, E72 and Q73) determined?

Response: We revised the *in silico* analyses section and focused on interactions with residues that were identified as potential interaction partners by data of *in vitro* experiments.

h. How was the secondary structure “abundance” calculated?

Response: In agreement with the comments of reviewer #1 we reduced the *in silico* analyses and focused it on the influence of ionic interactions between E72 and the surrounding residues. Therefore, secondary structure analyses were removed.

Reviewer #3 (Remarks to the Author):

In the revised manuscript the authors have addressed most of our previous comments, however, we still have concerns about the possible artifacts caused by immobilizing a part of their system in their MD simulations.

From the revised manuscript it is still unclear how the system was immobilized and how the boundary between mobile and immobile parts was treated in their MD simulations. Was the surrounding solvent immobilized together with the protein, and was the immobilized part of the system completely excluded from the computation? It's uncommon to immobilize part of the system in MD simulations as the authors have implemented. We would like the authors to reference works showing that such an approach does not introduce significant artifacts at least at the chosen 45Å cutoff.

We suggest the authors produce at least one full-length (100ns) simulation to demonstrate that the unconstrained MD simulations display RMSDs/RMSFs compared to their "local" simulations. Since the authors have already achieved 5ns of unrestrained simulation during equilibration, we believe that such validation could be performed in a feasible amount of time.

Finally, the authors need to specify how the RMSF was computed: which part of the structure was used for alignment and which atoms were to compute the RMSF.

Point-by-point response to the reviewer remarks

Response to reviewer #2

The authors have made a substantial effort to improve the quality of all pulldown assays in the revised manuscript, however, the interpretation of some of the data remains problematic. Specifically, in Figure 6a the authors observe reduced HA-PA signal in the pulldown using PB1-K578A, but the levels of PB1 and PB2 in the pulldown are equal to wild type. The authors interpret this result as a reduced NP-RdRp interaction, but if this were the case then PB1 and PB2 would also be reduced in the pulldown, which they are not. The fact that only PA is reduced in the pulldown instead indicates that the PB1 K578A mutation is preventing PA from interacting properly with PB1 and PB2; in other words, the mutation prevents the trimeric RdRp from assembling correctly.

RESPONSE: We thank the reviewer for the recognition and approval of the improvements that were made to the manuscript. We regret, that the improved results and our conclusion for Fig. 6a are not convincing to the reviewer. However, although we can follow the reviewer's argumentation, we disagree with the reviewer's interpretation of our data due to the following reasons:

1. The docking site of NP to the active trimeric viral polymerase, including the structurally distinct complexes of the transcriptase and replicase, has remained widely unresolved. While NP interaction sites have been mapped to the PB1 and PB2 subunits but not PA, the biologically relevant interaction site(s) in the trimeric enzyme are not known and were not yet structurally resolved.
2. The results of our IP in figure 6a and supplementary figure 4a exactly resemble these previous findings as we observe NP interaction with PB1 and PB2 but not the PA subunit when co-expressed individually.
3. Since PA lacks a direct interaction site to NP and needs to be co-precipitated with the other subunits, we chose this subunit for our experimental readout to report NP interaction with the trimeric polymerase. However, during co-expression, NP will interact with the trimeric polymerase as well as with soluble PB1 and PB2 molecules that were not incorporated. Hence, NP precipitation will also result in co-precipitation of soluble PB1 and PB2 subunits. However, westernblot analysis will not distinguish whether the precipitated subunits are derived from the trimeric complex or soluble protein. The levels of NP, PB1 and PB2 therefore do not correlate with the amount of

precipitated trimeric polymerase in our assay. This is only achieved by PA. However, we are also aware, that we cannot exclude or determine the amount of NP co-precipitation with dimers of PA/PB1 or PA/PB2.

4. The conclusion that PB1-K578A prevents PA from interacting with PB1 and PB2 and thus would lead to reduced assembly of the trimeric polymerase appears highly unlikely to us. Reduced assembly of the trimeric complex would also negatively affect the activity of the polymerase in the reconstitution assay. Instead, our data demonstrate that PB1-K578A enhances the polymerase activity (Fig. 3e).

Responses to Reviewer #3

In the revised manuscript the authors have addressed most of our previous comments, however, we still have concerns about the possible artifacts caused by immobilizing a part of their system in their MD simulations.

RESPONSE: MD simulations are *in silico* models, that cannot display reality in full detail, but can be utilized to form a working hypothesis on putative dynamic structural rearrangement. Consequently, artefacts can occur very easily based on many different aspects. Even the usage of homology models as a base for the simulations can lead to results, that are not in accordance with the *in vitro* or *in vivo* data. Therefore, it is essential to proof and confirm every *in silico* derived data by suitable and multi-facetted *in vitro* / *in vivo* experiments, which are also part of this manuscript.

We would like to point out that the *in silico* analysis was used to simulate local structural rearrangements which were suspected to occur upon ubiquitination-mediated charge neutralization of K578 in the PB1 subunit of the viral polymerase, without making further predictions on the global effects within this multimeric and highly complex molecule. This locally restricted approach was justified by a previous publication, which revealed that the interaction surface between two trimeric polymerases was situated in the nearest vicinity of PB1-K578, between the individual PA subunits of each polymerase trimer, under direct involvement of several neighboring amino acids. The results of the *in silico* analysis indeed pointed into the direction that not only the linkage of a bulky ubiquitin moiety affected the polymerase dimerization, but also suggested that charge-mediated structural rearrangements at the interaction surface between the polymerases could be involved, which was subsequently further supported by the results of our *in vitro* experiments.

While we appreciate the reviewers concerns about possible artifacts in the *in silico* analysis, we would like to point out that in combination, the *in silico* analysis and our *in vitro* data provide a plausible and biologically relevant mechanistic model of how IAV utilize host derived PTMs to control the functions of the viral polymerase during the course of infection. We would further like to draw the attention of the reviewer to the most important results of our comprehensive and state of the art experimental analysis to highlight the robustness and remarkable alignment of the *in silico* prediction with the experimental data:

- Residue K578 in subunit PB1 of the viral polymerase undergoes ubiquitination by a cellular E3 ligase during infection.
- The first experimental evidence for a charge mediated effect through PB1-K578 ubiquitination derived from the polymerase reconstitution assay in which a fully functional IAV polymerase is reconstituted from plasmids in transfected cells (Fig. 2e). Both A and R substitutions of residue PB1-K578 resulted in an increased polymerase activity, reflected by increased reporter gene expression, suggesting that the charge at K578 affects the polymerase function and that both charge states play non-redundant roles for viral replication.
- Furthermore, the R mutation did not support reconstitution of a stable recombinant virus, which provided strong evidence for an incompatibility of K578R at a certain step of viral replication, despite the positive effect on the polymerase activity in the previous experiment. Instead, we observed that K578R was under strong mutation pressure and reverted to the natural K (Fig. 5a and b), providing solid proof for its high biological relevance, most likely because of its susceptibility to ubiquitylation. This result supported the previous finding that, both, the positively charged (non-modified) and neutral (ubiquitinated) states of K578 are required for optimal viral replication.
- From published crystal structures, we developed the idea that PB1-K578 could be engaged in a charge-mediated interaction to loop residue PB2-E72, which was strongly supported by mutation of this residue to E72A, which also resulted in the upregulation of the polymerase activity (Fig. S3h and i). This is also supported by the *in silico* analysis, which supports that the interaction between PB1-K578 and PB2-E72 is stabilized in a charge dependent manner under involvement of several neighboring residues.
- Most importantly, our interaction studies with mutated proteins confirmed that dimerization of the viral polymerase via the neighboring PA/PA interface is affected in a charge-dependent manner (Fig. 6c), which provided compelling *in vitro* evidence for the occurrence of structural rearrangements at the interaction site that are conferred by the charge state of residue PB1-K578. Here, our data suggest that a neutral charge at PB1-K578 mediated by Ub linkage reduces the affinity for the assembly of the

polymerase dimer, which is required to provide the optimal equilibrium of monomeric to dimeric polymerases, while a constitutive positive charge shifts this equilibrium towards the dimer which is not compatible with viral replication.

- In summary, our *in vitro* data support our initial hypothesis that locally restricted structural rearrangements conferred by ubiquitination of PB1-K578 rather than changes in distant regions fine tune the interaction between two viral polymerases at the PA/PA interface.

Since all results of our MD simulations are in clear accordance with the conducted *in vitro* experiments as pointed out here, we are convinced that no major artefacts occurred, that would substantially change the outcome and conclusion of our study.

While we cannot exclude that charge changes at residue PB1-K578 also confer structural changes at other, more distant, parts of the polymerase, we estimate the relevance of these alterations for the observed changes in polymerase dimerization at the PA/PA interface to be very low and therefore not within the scope of this manuscript. Finally, we would like to emphasize that the interpretation of our results and the final model are closely and carefully aligned to the current advances in the understanding of the structure-function relationship within the viral polymerase during viral mRNA transcription and genome replication, which are derived from bona fide crystal structures.

We hope that this comprehensive summary convinces Reviewer #3 that the results of the *in silico* analysis are well supported by our extensive *in vitro* results.

From the revised manuscript it is still unclear how the system was immobilized and how the boundary between mobile and immobile parts was treated in their MD simulations. Was the surrounding solvent immobilized together with the protein, and was the immobilized part of the system completely excluded from the computation?

RESPONSE: We specified the procedure in materials and methods again and added additional explanations (blue text):

The homology model of the three-dimensional structure of the WSN polymerase complex bound to cRNA was used for further structural analyses. The residue PB1-K578 was mutated to K578A or K578R leading to three independent models for further simulations. Each model was prepared and energy minimized for local simulations by initialization procedure of YASARA structure (version 21.8.27)⁹¹ using AMBER14⁹² force field including structure cleaning and hydrogen network optimization^{93,94}, generation of a water shell (TIP3P) around the protein model as well as prediction of pKA values at the chosen pH of 7.4⁹⁵. After short

equilibration simulation (~5 ns, AMBER14) of the whole protein, the models were subjected to 100 ns local molecular dynamic simulations. Since previous results clearly identified the PB1-PB2 loop-interface as the crucial region for altered behavior of K578R and K578A, molecular dynamic (MD) simulations were conducted to evaluate the interplay between these two domains. To achieve a simulation time sufficient for interplay analyses, local MD simulations were performed, which were only possible due to the strong separation of single functional domains in the protein complex. Especially the PB1-PB2 loop-interface can be identified as an independent domain of the complex, that encompasses a spatial dimension equal to a 45Å-sphere. Therefore, atoms in a 45Å-sphere around residue PB1-K578 including only protein residues were set mobile, while all other atoms were immobilized. Further, the size of the simulation box was limited to 92.95 x 92.95 x 92.95 Å, which is sufficient to include a fully mobile water shell, all mobile protein residues as well as enough immobilized protein residues to ensure stability and integrity of the protein. On the other hand, this procedure reduces the possibility of extensive MD artefact formation due to the presence of large amounts of immobilized residues. Using this approach, we optimized the needed calculation time and computational resources leading to a simulation time of 100 ns for each replicate, which was not achievable in an all-atoms mobile simulation for the complete protein. Local MD simulations were conducted using the following settings: AMBER15IPQ force field⁹⁶, particle-mesh Ewald / Poisson-Boltzmann cutoff 8Å, periodic simulation cell boundary, long range coulomb forces, 0.9% NaCl, pH=7.4, water density 0.997 (TIP3P), pressure of 1 atm and a simulation temperature of 298 K. Complete simulation including temperature and pressure settings was controlled by a NPT ensemble and atomic motions were integrated with a multiple timestep of 2x1.25 fs for bonded interactions and 2.5 fs for non-bonded interactions as previously described⁹⁷. Local simulations were replicated to n=4 for all conditions (WT, K578R, K578A). Each simulation was documented by simulation snapshots every 0.1 ns leading to a total number of 4000 analyzable snapshots for each condition.

It's uncommon to immobilize part of the system in MD simulations as the authors have implemented. We would like the authors to reference works showing that such an approach does not introduce significant artifacts at least at the chosen 45Å cutoff.

RESPONSE: We agree with the reviewer, that this approach is unusual. However, similar assumptions and simplifications were made in different publications. Many MD simulations are based on crystal or cryo-EM structures, that do not cover the complete sequence of a protein. For example, simulations using structures of the NMDA (Durham et al 2020; PDB 4PE5) do not include the C terminal domain, which is strongly involved in channel function (Sapkota et al 2019). Also, for other ion channels like KCNQ1 MD simulations of isolated domains like the voltage sensor domain and pore region were performed, while other parts of the protein were

excluded (Willegems et al 2022). Moreover, limiting the MD simulations to structural isolated domains is also known in the field of virus proteins. For example, Alshawaf et al performed MD simulations to examine the potential of 3-O-methylquercetin to act as an inhibitor of protein-protein interactions between neuropilin-1 and the S-protein of the SARS-CoV-2 virus (Alshawaf et al. 2022). The performed MD simulations only included the tested compound and the neuropilin-1 protein, but not the S protein. Therefore, it is quite common to simulate isolated domains of ion channels or virus proteins. In our case, the procedure was chosen because of the strong spatial isolation of the examined region of interest (see Figure 1). We are clearly aware of the risks that are associated with these kinds of restrictions in MD simulations. Therefore, we used all *in silico* data as a visualization but not as a clear proof of the hypothesis.

Figure 1: PB1-PB2 complex (gray) with the simulated domain (yellow). Residues in a distance of $<10 \text{ \AA}$ to the loop residue PB2 E72 are colored in purple.

We suggest the authors produce at least one full-length (100ns) simulation to demonstrate that the unconstrained MD simulations display RMSDs/RMSFs compared to their “local” simulations. Since the authors have already achieved 5ns of unrestrained simulation during equilibration, we believe that such validation could be performed in a feasible amount of time.

RESPONSE: We would like to point out to the Reviewer that all results are in clear accordance with the *in vitro* data set as described above. Therefore, we do not agree with the Reviewer and strongly question that the overall outcome of our study would benefit from an extended validation. In contrast, we are concerned that an extended analysis of the predicted global structural changes will shift the focus to non-relevant structural changes that do not stand in direct association to the proven ubiquitination of residue PB1-K578 and are therefore beyond the scope of our manuscript. Furthermore, a single simulation would not generate any statistically relevant data, that could identify substantial and relevant artefacts. The lack of statistically evaluated data were previously criticized by the reviewer.

Finally, the authors need to specify how the RMSF was computed: which part of the structure was used for alignment and which atoms were to compute the RMSF.

RESPONSE: The root mean square fluctuation (RMSF) is a parameter to describe mobility of distinct atoms or residues in a protein over the complete simulation time. To specify the calculation of the RMSF values we added the following paragraph to the materials and methods section:

The calculation of the RMSF is performed in three automated steps. First, the mean position ($\overline{P_{ij}}$) of the distinct atom (i) is calculated using atom position vector (P) with j=3 components for the x, y and z coordinates of every single snapshot (k) from the 100 ns simulation (N=1000). In the second step, the root mean square fluctuation is calculated by the following equation:

$$RMSF_i = \sqrt{\sum_{j=1}^3 \left(\frac{1}{N} \sum_{k=1}^N P_{ikj}^2 - \overline{P_{ij}}^2 \right)}$$

Both steps are performed for all atoms individually. In the final step, the RMSF for each residue is calculated by the average RMSF of its constituting atoms.

REFERENCES

Durham, R. J., Paudyal, N., Carrillo, E., Bhatia, N. K., Maclean, D. M., Berka, V., Dolino, D. M., Gorfe, A. A., & Jayaraman, V. (2020). Conformational spread and dynamics in allostery of NMDA receptors. *Proceedings of the National Academy of Sciences of the United States of America*, 117(7), 3839–3847. <https://doi.org/10.1073/pnas.1910950117>

Sapkota, K., Dore, K., Tang, K., Irvine, M., Fang, G., Burnell, E. S., Malinow, R., Jane, D. E., & Monaghan, D. T. (2019). The NMDA receptor intracellular C-terminal domains reciprocally interact with allosteric modulators. *Biochemical pharmacology*, *159*, 140–153. <https://doi.org/10.1016/j.bcp.2018.11.018>

Willegems, K., Eldstrom, J., Kyriakis, E. et al. Structural and electrophysiological basis for the modulation of KCNQ1 channel currents by ML277. *Nat Commun* 13, 3760 (2022). <https://doi.org/10.1038/s41467-022-31526-7>

Alshawaf, E., Hammad, M. M., Marafie, S. K., Ali, H., Al-Mulla, F., Abubaker, J., & Mohammad, A. (2022). Discovery of natural products to block SARS-CoV-2 S-protein interaction with Neuropilin-1 receptor: A molecular dynamics simulation approach. *Microbial pathogenesis*, *170*, 105701. <https://doi.org/10.1016/j.micpath.2022.105701>

REVIEWERS' COMMENTS

Reviewer #3 (Remarks to the Author):

On one hand, I am not fully convinced by the justifications provided by the authors on not conducting the validations using the full-length MD simulations. As the authors pointed out themselves, their "local" simulations could introduce artifacts. We did a quick back-of-envelope calculation that even a regular desktop computer (e.g., 8 threads on an i5 CPU) can achieve ~10ns/day for a system containing 100K atoms. With a single GPU, one can easily run over 100ns/day for this system. In my opinion, running full-length MD simulations seems a task that can be achieved within a reasonable amount of time.

On the other hand, I also agree with the authors that the observations from their "local" MD simulations are fully consistent with their experimental results.

Even if the authors only aimed to use MD simulations for "visualization" but not as a clear proof of the hypothesis, and everything they see in MD does not contradict their experiment. In my own opinion, I still think that a rigorous way of performing MD simulations needs to be utilized.

Apart from the above point, the authors have appropriately addressed my other concerns.

Point-by-Point response to the Reviewer

Reviewer #3 (Remarks to the Author):

On one hand, I am not fully convinced by the justifications provided by the authors on not conducting the validations using the full-length MD simulations. As the authors pointed out themselves, their "local" simulations could introduce artifacts. We did a quick back-of-envelope calculation that even a regular desktop computer (e.g., 8 threads on an i5 CPU) can achieve ~10ns/day for a system containing 100K atoms. With a single GPU, one can easily run over 100ns/day for this system. In my opinion, running full-length MD simulations seems a task that can be achieved within a reasonable amount of time.

On the other hand, I also agree with the authors that the observations from their "local" MD simulations are fully consistent with their experimental results.

Even if the authors only aimed to use MD simulations for "visualization" but not as a clear proof of the hypothesis, and everything they see in MD does not contradict their experiment. In my own opinion, I still think that a rigorous way of performing MD simulations needs to be utilized.

Apart from the above point, the authors have appropriately addressed my other concerns.

Response:

We thank the reviewer for highlighting the consistency of the MD and with our experimental data. Nevertheless, we also appreciate the remaining positive criticism of Reviewer #3 which has already supported us to improve the quality and robustness of our manuscript in the last revisions and which we believe is vital for scientific progression.

We feel that we have now addressed the shortcomings of the applied dynamic structural simulation technique clearly in our manuscript and also highlighted its hypothesis building character and we are convinced that the relevance and limitations of this analysis will now be correctly understood by the readership.